# Unleashing the Potential of Unlabeled Data: Bidirectional Collaborative Semi-Supervised Active Learning for 3D Object Detection

## Abstract

To address the annotation burden in LiDAR-based 3D object detection, active learning (AL) methods offer a promising solution. However, traditional active learning approaches solely rely on a small amount of labeled data to train an initial model for data selection, overlooking the potential of leveraging the abundance of unlabeled data. Recently, attempts to integrate semi-supervised learning (SSL) into AL with the goal of leveraging unlabeled data have faced challenges in effectively resolving the conflict between the two paradigms, resulting in less satisfactory performance. To tackle this conflict, we propose a **B**idirectional **C**ollaborative **S**emi-**S**upervised **A**ctive **L**earning framework, dubbed as BC-SSAL. Specifically, from the perspective of SSL, we propose a Collaborative PseudoScene Pre-training (CPSP) method that effectively learns from unlabeled data without introducing adverse effects. From the perspective of AL, we design a Collaborative Active Learning (CAL) method tailored for outdoor LiDAR scenes, which complements the uncertainty and diversity methods by model cascading, alleviating the dilemma of sampling rare classes. Extensive experiments conducted on KITTI and Waymo demonstrate the effectiveness of our BC-SSAL. Especially, on the KITTI dataset, utilizing only 2% labeled data, BC-SSAL can achieve comparable performance to the model trained on the full set.

## 1 Introduction

Being a fundamental task in autonomous driving, LiDAR-based 3D object detection plays a crucial role in perceiving semantic and spatial clues, which recognizes and locates objects in 3D scenes based on input point clouds captured by LiDAR sensors. During the past few years, a large number of efforts (Yan et al., 2018; Zhou & Tuzel, 2018; Lang et al., 2019; Chen et al., 2022; Yang et al., 2020) have been made with the performance of major public benchmarks (Geiger et al., 2013; Caesar et al., 2020; Sun et al., 2020) rapidly and consistently increasing. Unfortunately, current methods are deep learning based, substantially dependent on labeled data. For instance, the Waymo dataset (Sun et al., 2020) alone encompasses over 10 million ground-truth (GT) 3D boxes. The labor-intensive and time-consuming nature of annotating extensive datasets creates a bottleneck, hindering the advancement in this field.

Active learning (AL) (Haussmann et al., 2020; Li et al., 2021; Feng et al., 2019) offers a promising solution to overcoming this drawback. It selects a small subset from all samples as the most informative data to measure the benefits of a fully annotated dataset. By adaptively choosing "good" samples to label, AL significantly reduces the burden of data acquisition and annotation and shows the potential to facilitate LiDAR-based 3D object detection (Luo et al., 2023b;a; Jiang et al., 2022; Schmidt et al., 2020).

The AL paradigm typically consists of three phases, *i.e.* (1) temporary model updating (TMU), (2) unlabeled sample selecting (USS), and (3) final model delivering (FMD). In TMU, a temporary model is built or enhanced with the set of available labeled data, which is further applied to generate pseudo annotations; in USS, some unlabeled data are screened out according to certain criteria and annotated by the temporary model obtained; and in FMD, the final model is output. In general, TMU and USS are jointly conducted for multiple iterations while FMD operates once in the end.

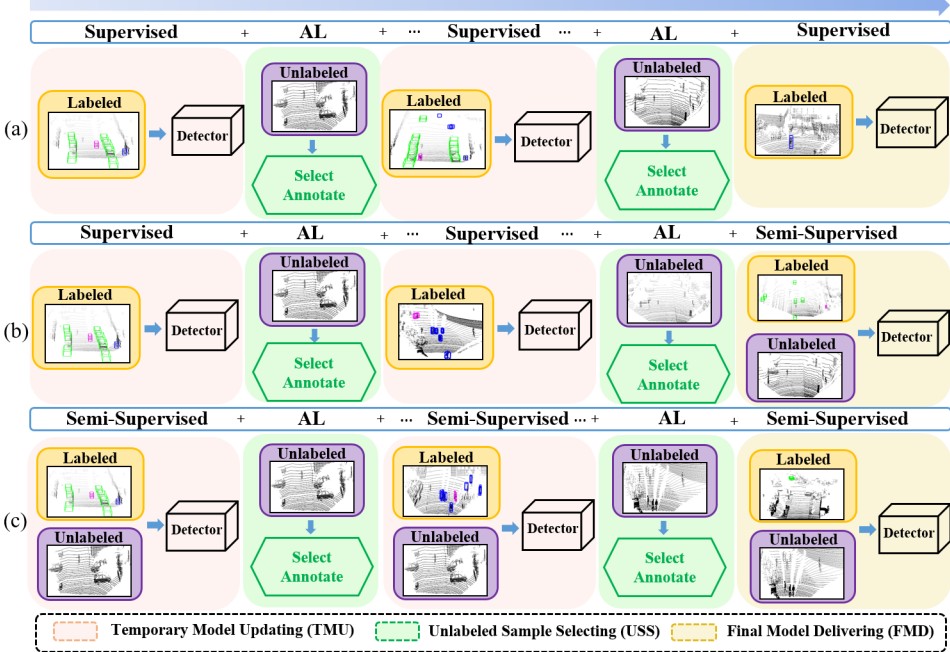

Figure 1: The illustration depicts different paradigms for combining Active Learning (AL) and Supervised/Semi-supervised Learning (SL/SSL): (a) Solely utilizing SL in all stages. (b) Employing SSL only in the final model delivering stage. (c) Integrating SSL across all stages. Paradigm (c) achieves enhanced performance by incorporating unlabeled data compared to paradigm (a). However, traditional SSL methods face conflicts with AL in the temporary model updating stage, leading to suboptimal data selection. Thus, paradigm (c) performs less effectively than paradigm (b).

The traditional AL methods only make use of labeled data, as shown in Fig. 1 (a). Since the large amount of unlabeled data conveys rich information, which helps better understand the distribution of all data rather than that of labeled ones, overlooking them leaves much room for improvement.

With the progress achieved in semi-supervised learning (SSL), some preliminary attempts (Lyu et al., 2023; Wang et al., 2023; Mi et al., 2022) have been made to integrate such techniques in AL, where SSL contributes to the performance gain by assigning pseudo-labels to unlabeled data based on the prediction of the model trained on label data (Zhao et al., 2020; Wang et al., 2021). As depicted in Fig. 1 (c), they employ SSL to strengthen both the temporary and final model in the TMU and USS phases and demonstrate that the semi-supervised active learning (SSAL) paradigm is superior to the traditional one (Fig. 1 (a)). However, the combination of SSL and AL is not as straightforward as they expect, in particular for the synergy of TMU and USS. As we know, uncertainty-based metrics are widely adopted in AL and the samples with higher uncertainties are more likely to be selected for annotation in USS. On the other side, the samples of higher uncertainties may suffer from low confidence scores due to the instability of SSL in TMU. In this case, a conflict arises, where the assignment of incorrect pseudo labels to objects in SSL inevitably becomes a significant source of noise and makes AL struggle to accurately assess their uncertainties. As this iterates for multiple rounds, current SSAL is prone to converge with sub-optimal results, even inferior to that of a degraded paradigm only applying SSL in FMD (Fig. 1 (b)).

To tackle the conflict between SSL and AL, we propose a bidirectional collaborative semi-supervised active learning (BC-SSAL) framework. **From the perspective of SSL**, in TMU, we present a method, namely collaborative pseudo-scene pre-training (CPSP), to effectively leverage unlabeled data while bypassing the side effects aforementioned. The main idea is to selectively learn only from confident objects. To this end, we generate pseudo-scenes of unlabeled data using confident objects with the ones of high uncertainties excluded. Pre-training on these pseudo-scenes thus ensures that unconfident objects are not disturbed by their own pseudo-labels, which largely mitigates the negative impact of mislabeling and noise. **From the perspective of AL**, considering that objects in outdoor LiDAR scenes show a severely long-tailed distribution and many objects of different

classes appear in the same scene, it is quite difficult for AL to sample objects of rare classes for annotation. Moreover, accurately counting objects is challenging, as some background regions may be misclassified as unconfident objects, further complicating the balanced sampling of scenes in active learning. Therefore, we design a collaborative active learning (CAL) method, which simultaneously takes the uncertainty and diversity into account. In contrast to previous counterparts, CAL enhances uncertainty estimation and class weighting by cascading the model trained on labeled data with the one trained on labeled and unlabeled data. Finally, BC-SSAL collaboratively integrates AL and SSL by bidirectionally adapting them to fit each other.

In summary, our contributions are as follows:

- We point out the conflict between AL and SSL and propose a novel SSAL framework to address it, where the CPSP method is presented to effectively leverage unlabeled data to facilitate model training.
- We design the CAL method tailored for outdoor LiDAR scenes, which complements the uncertainty and diversity methods by model cascading, alleviating the dilemma of sampling rare classes.
- We do extensive experiments on the KITTI and Waymo datasets and reach state-of-the-art results. Especially, on KITTI, we use only 2% labeled data and achieve comparable performance to the model trained on the full set.

## 2 RELATED WORK

### 2.1 ACTIVE LEARNING

Active learning methods have gained significant attention in various domains to alleviate the labeling burden. These methods can be broadly categorized into two main types: uncertainty-based (Houlsby et al., 2011; Gal et al., 2017) and diversity-based approaches (Nguyen & Smeulders, 2004; Sener & Savarese, 2017; Agarwal et al., 2020). Uncertainty-based methods leverage uncertainty to identify informative samples for annotation while diversity-based methods prioritize capturing the diversity and representativeness of the dataset. Furthermore, recent research (Huang et al., 2010; Ash et al., 2019) has explored the integration of uncertainty-based and diversity-based approaches to leverage the advantages of both.

Recently, there has been increased interest in applying active learning to object detection tasks. Unlike image classification, active learning for object detection presents unique challenges due to the complexities of localizing and identifying objects within images. One approach, MI-AOD (Yuan et al., 2021) treats unlabeled images as bags of instances, using adversarial classifiers to measure uncertainty. AL-MDN (Choi et al., 2021) utilizes mixture density networks for probabilistic outputs, while ENMS (Wu et al., 2022) applies entropy-based non-maximum suppression to assess uncertainty. PPAL (Yang et al., 2022) offers a plug-and-play active learning method.

However, active learning for LiDAR-based object detection needs further research due to the differences between images and outdoor LiDAR scenes. Some recent studies have begun to tackle this issue. For example, CRB (Luo et al., 2023b) focuses on filtering redundant 3D bounding box labels based on conciseness, representativeness, and geometric balance. KECOR (Luo et al., 2023a) presents a novel strategy called kernel coding rate maximization to identify the most informative point clouds for labeling. However, these methods may struggle with class imbalance caused by long-tailed distributions in outdoor scenes and do not effectively utilize available unlabeled data.

### 2.2 SEMI-SUPERVISED ACTIVE LEARNING

Semi-supervised learning (SSL) techniques (Xu et al., 2021; Liu et al., 2021; Zhao et al., 2020; Wang et al., 2021; Yin et al., 2022; Liu et al., 2023; Gao et al., 2023) aim to enhance model performance by leveraging abundant unlabeled data. These methods can be integrated with active learning (AL) to further optimize data annotation efforts (Elezi et al., 2022; Mi et al., 2022; Lyu et al., 2023; Wang et al., 2023; Hwang et al., 2023). In most frameworks, SSL is employed for model pre-training during the Temporary Model Updating (TMU) stage, after which AL identifies the most informative samples for annotation. However, many approaches overlook the potential conflicts between SSL

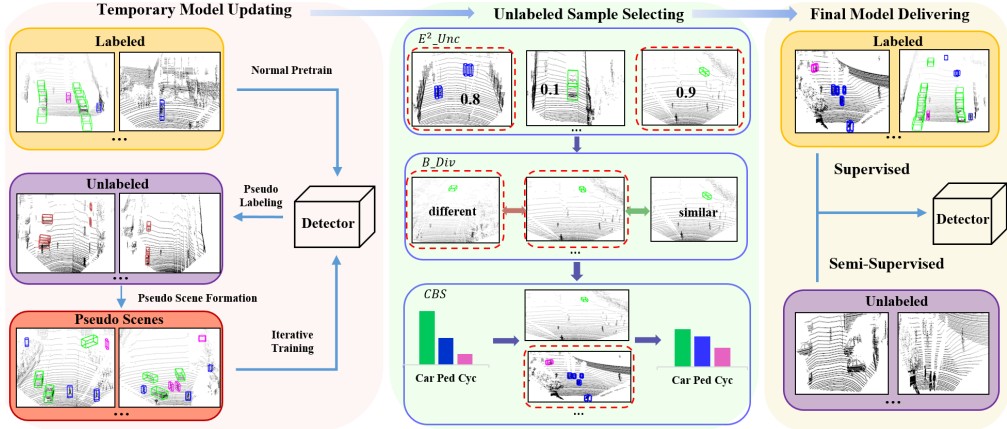

Figure 2: Overview of our BC-SSAL framework. In the Temporary Model Updating stage(TMU), we propose creating pseudo scenes with confident objects for model pre-training (CPSP). Subsequently, in the Unlabeled Sample Selecting stage(USS), we design a collaborative active learning method to select valuable data for annotation (CAL). Finally, in the Final Model Delivering stage(FMD), we leverage traditional semi-supervised learning methods to enhance the model performance.

and AL during TMU. These methods often rely on pseudo-labeling techniques that may degrade performance due to the noises (Mi et al., 2022; Lyu et al., 2023; Wang et al., 2023). On the other hand, (Hwang et al., 2023) mainly uses consistency loss, which is less affected by conflicts between SSL and AL, but it still lacks sufficient support for effective semi-supervised learning in 3D object detection. Similarly, (Elezi et al., 2022) uses an auto-labeling scheme to reduce distribution drift. However, this method relies on a specific loss function, making it difficult to supervise established 3D detectors. As a result, it may struggle to learn effectively from unlabeled data in 3D detection tasks. In this paper, we propose a bidirectional collaborative semi-supervised active learning framework, which addresses the conflicts between SSL and AL, effectively unleashing the potential of unlabeled data for 3D object detection.

## 3 METHOD

### 3.1 FRAMEWORK OVERVIEW

As illustrated in Fig. 2, our **B**idirectional **C**ollaborative **S**emi-**S**upervised **A**ctive **L**earning framework (BC-SSAL) consists of three main components: Temporary Model Updating (TMU), Unlabeled Sample Selecting (USS), and Final Model Delivering (FMD). **In the TMU stage**, we initiate the process with normal pre-training, where a small set of randomly sampled data is used to train the initial model. Subsequently, our Collaborative PseudoScene Pre-training tailored for active learning is performed, creating pseudo scenes with confident boxes to enhance the model performance. **In the USS stage**, we employ the innovative Collaborative Active Learning method, which entails the strategic selection of informative data from the unlabeled pool, empowering the model to concentrate on challenging instances. **In the FMD stage**, semi-supervised learning is conducted to further refine the model performance, which utilizes both labeled and unlabeled data to train the model, capitalizing on insights gained from the active learning process. It's important to note that our framework is compatible with various existing semi-supervised methods, providing flexibility in choosing the most suitable approach.

### 3.2 COLLABORATIVE PSEUDOSCENE PRE-TRAINING

To meet the requirements of active learning, we propose the Collaborative PseudoScene Pre-training(CPSP) approach, which is specifically designed to support active learning by creating pseudo scenes that focus on confident objects while excluding unconfident boxes. The entire process is illustrated in Fig. 3.

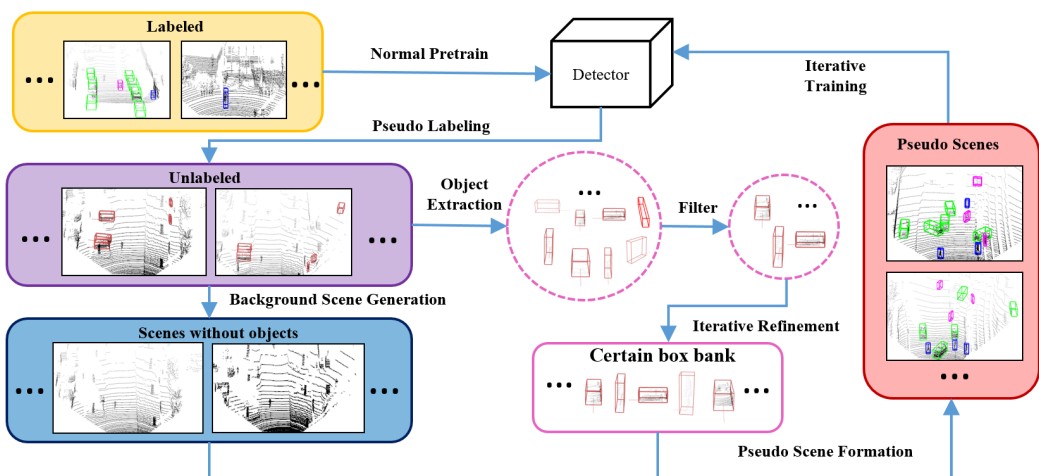

Figure 3: The illustration of the Collaborative PseudoScene Pre-training (CPSP) module. We extract confident objects from unlabeled scenes based on their uncertainty and store them in a box bank, which is iteratively updated to maintain its quality. Additionally, we remove point clouds corresponding to the predicted boxes, creating "background" scenes without any objects. The point cloud from the box bank is then inserted into these "background" scenes, forming pseudo scenes.

### 3.2.1 CONFIDENT OBJECT EXTRACTION

To optimize the extraction of confident objects for model pre-training, we employ a multi-step approach that enhances the quality of our training data. Initially, we utilize a Confident Object Filtering module to extract confident objects from unlabeled scenes, providing crucial information for model training. These extracted objects are then stored in a box bank to preserve and manage the collected object information. Furthermore, we incorporate an Iterative Refinement mechanism that iteratively generates confident boxes from the unlabeled data, integrating them into the box bank to create high-quality pseudo labels for the models. This process is essential for improving the robustness and accuracy of the model. Please see appendix B.1 for more details.

**Confident Object Filtering.** To ensure compatibility with active learning(AL), we utilize the same uncertainty measure employed in the AL process. By applying this uncertainty measure to the objects, we collect their uncertainty scores and employ clustering techniques on these scores to identify the group with the lowest uncertainty scores. After that, we filter the objects within this group to select confident objects. The object information is represented as $O = \{cls, loc, score, scene_{id}, pc\}$, where $cls$ denotes the class labels, $loc \in R^7$ represents the object location with orientation, $score \in R^1$ indicates the uncertainty score, $scene_{id}$ corresponds to the scene index to which the object belongs, and $pc \in R^{n \times 3}$ captures the point clouds of the object. Additionally, we extract some backgrounds that are likely to be false positives, similar to the approach in (Oh et al., 2024). All extracted object information is stored in a box bank for easy access and management.

**Iterative Refinement** To continuously improve the box bank, we implement a mechanism that adds newly extracted confident objects. When a new object overlaps with an existing one, its uncertainty scores are compared, and the object with the lower score is retained, ensuring that only higher-confidence objects are prioritized for further processing. However, even with low uncertainty scores, errors may still occur, potentially impacting uncertainty estimation during active learning. To mitigate this, we introduce a deletion mechanism: after each training iteration and the extraction of new confident boxes, the newly added boxes are compared with the existing ones to identify and remove any mislabeled or erroneous entries. This iterative process refines the box bank, improving its overall quality and ensuring more reliable uncertainty estimation.

### 3.2.2 PSEUDO SCENE FORMATION

To enhance the model's focus on confident objects while minimizing the impact of unconfident ones, we utilize the Reliable Background Mining Module from (Liu et al., 2022). Moreover, we establish a relatively high threshold to preserve backgrounds prone to being falsely identified as positives. This module effectively removes point clouds linked to predicted boxes from unlabeled scenes,

accounting for the sparse nature of point cloud data. By leveraging these "background" scenes, we create a contextual learning environment for the model to concentrate on relevant objects. Next, we construct Pseudo Scenes by merging point clouds from selected boxes in the box bank with these background scenes. These Pseudo Scenes consist solely of confident objects, excluding any unconfident ones from the training data. This strategy ensures that the pre-training data is tailored to enhance the model's ability to make stable and reliable predictions, providing a robust foundation for the active learning process.

### 3.3 COLLABORATIVE ACTIVE LEARNING

To efficiently identify the most informative samples and achieve better collaboration with the semi-supervised pre-training stage, we propose a novel Collaborative Active Learning (CAL) approach, which simultaneously incorporates considerations of uncertainty and diversity. For uncertainty, we devise Ensemble-based Entropy Uncertainty ($E^2\_Unc$). In terms of diversity, our approach includes Box-level Diversity ($B\_Div$) and Class Balance Sampling (CBS). A more detailed algorithm is provided in Appendix B.2.

#### 3.3.1 ENSEMBLE-BASED ENTROPY UNCERTAINTY

We use entropy to measure the uncertainty of each predicted box. Considering the collective influence of all the boxes, the overall uncertainty of the entire scene is represented by calculating the average entropy. This approach enables us to capture the overall uncertainty and make informed decisions based on the entropy measure.

Specifically, the uncertainty for a point cloud scene $S$ is computed as:

$$H(S) = \frac{\sum_{b \in S} \sum_{c \in C} (-p_{bc} \log p_{bc})}{N_b \times |C|} \tag{1}$$

where $b$ represents the predicted boxes, $p_{bc}$ is the predicted class probability of class $c$ for box $b$, $N_b$ is the total number of predicted boxes, and $|C|$ is the number of object classes.

We observe that the CPSP pre-trained model can overlook some original correct and confident objects. To address this, we propose an ensemble strategy that combines high-confidence predictions from the normal pre-trained model with all boxes from our CPSP pre-trained model. We then apply the Non-Maximum Suppression (NMS) technique to eliminate redundant boxes.

#### 3.3.2 BOX-LEVEL DIVERSITY

Diversity is essential for reducing redundancy in the selected samples. We achieve this by measuring the similarity between boxes using cosine similarity and assigning each box to its closest counterpart. The similarity score for each scene is computed by averaging the cosine similarity between box features from the current scene and features from previously selected scenes.

Formally, let $S_a$ be a scene with box features $F_a = \{f_{a,i} \mid \text{box}_{a_i} \in S_a\}$, and $S = \{S_c\}$ be the set of selected scenes with features $F = \{f_i \mid \text{box}_i \in S_c\}$. We calculate the similarity of scene $S_a$ as:

$$Sim_a = \frac{1}{|F_a|} \sum_{i=1}^{|F_a|} \max_j \left( \frac{f_{a,i} \cdot f_j}{||f_{a,i}|| \cdot ||f_j||} \right) \tag{2}$$

During the sample selection process, if the similarity score between a new sample and previously selected samples exceeds a threshold, the sample is excluded from selection to avoid redundancy. Given the potentially large size of $|F|$, we apply clustering to retain the most representative features.

#### 3.3.3 CLASS BALANCE SAMPLING

Outdoor LiDAR scenes present significant challenges due to the presence of rare classes, which are difficult to sample and annotate. Annotating these rare classes becomes disproportionately expensive due to their limited representation, especially when they coexist with more frequent classes in the same scene. Additionally, many existing methods (Luo et al., 2023b; Wu et al., 2022; Yang et al., 2022; Luo et al., 2023a) fail to account for the fact that models often struggle to accurately estimate the number of objects in complex outdoor scenes, leading to a higher rate of false positives (FP) and false negatives (FN). To address this, we only consider boxes that are predicted by both the normal

pre-trained model and the CPSP pre-trained model. This intersection is more likely to represent real objects, filtering out background noise and reducing false positives.

Moreover, to handle class imbalance, we propose a class balance algorithm that sets an upper limit for the number of objects per class. If the number of objects in a class exceeds this limit, the weight assigned to that class is reduced. This encourages the sampling of unconfident objects from other classes in subsequent iterations, addressing the issue of class imbalance. As a result, our algorithm ensures more effective and representative sampling, improving the overall training process. More details are provided in Appendix B.2.

# 4 EXPERIMENTS

## 4.1 DATASETS AND IMPLEMENTATION DETAILS

### 4.1.1 KITTI DATASET

We conducted evaluations of our methods on the KITTI 3D detection benchmark (Geiger et al., 2013), using the standard train split comprising of 3,712 samples and the validation split containing 3,769 samples (Shi et al., 2020). In our semi-supervised active learning framework, we initially trained an initial model using randomly selected frames consisting of approximately 200 boxes. Subsequently, we leveraged the remaining unlabeled training data for further model refinement. During active learning, we specifically selected frames that contained around 150 boxes for effective training. The total number of labeled boxes in our approach is approximately 350 boxes, which accounts for less than 2% of the total boxes present in the KITTI train split. Additionally, we excluded scenes with "Dontcares" as they may introduce noise and potentially affect the performance of active learning methods. For evaluation, we calculate the mean average precision (mAP) at 40 recall positions for the Car, Pedestrian (Ped), and Cyclist (Cyc), employing 3D IoU thresholds of 0.7, 0.5, and 0.5, respectively, across different difficulty levels: easy, moderate (mod), and hard.

### 4.1.2 WAYMO DATASET

We conducted evaluations of our methods on the Waymo dataset (Sun et al., 2020), a widely used benchmark in autonomous driving. It offers diverse real-world driving scenarios with high-resolution sensor data. The dataset comprises 798 training sequences and 202 validation sequences. Notably, the annotations provide a full 360° field of view. Additionally, the prediction results are categorized into LEVEL 1 and LEVEL 2 for 3D objects based on the presence of more than five LiDAR points and one LiDAR point, respectively. To optimize efficiency, we adopted a time-saving approach by setting a sample interval of 20 from the training set to generate a pool of frames. From this pool, we selected frames for our divided datasets. Similar to our approach in the KITTI dataset, we employed a similar strategy for the Waymo dataset. In the initial stage, we randomly sampled frames with approximately 5000 boxes, and in the active learning stage, we again selected frames with around 5000 boxes. The total number of boxes, which amounts to 10,000, is less than 1% of the total boxes present in the Waymo train set. For evaluation, we use mean average precision (mAP) for Vehicle (Veh), Pedestrian (Ped), and Cyclist (Cyc) in LEVEL_1 (L1) and LEVEL_2 (L2), along with average mAP and heading accuracy weighted AP (mAPH).

### 4.1.3 IMPLEMENTATION DETAILS

As stated in (Lyu et al., 2023), the performance of object detection is closely tied to the number of boxes. To ensure a fair comparison with other methods, we maintain a fixed number of boxes rather than frames as the basis for our comparisons. In our implementation, we utilize PV-RCNN (Shi et al., 2020), a well-known model in active learning and semi-supervised learning, as our detector for the semi-supervised active learning framework.

In the stage of temporary model updating, we employ the same initial labeled and unlabeled data for all methods. This consistent approach allows for a fair comparison of the model when leveraging pre-training. Furthermore, in the final model delivering stage, we randomly initialize the model to assess the performance improvements achieved by selecting better data during the active learning procedure. Please see appendix C for more details.

Table 1: Comparison of results for various methods under different settings on the KITTI dataset. To ensure a fair comparison, we ensure that all frameworks utilize an identical amount of labeled data. Here, $N_1$ represents the initial box count, while $N_2$ signifies boxes selected through AL.

| Setting | $N_1/N_2$ | Pre-train | AL | SSL | Car_mod mAP | Ped_mod mAP | Cyc_mod mAP | Avg_easy mAP | Avg_mod mAP | Avg_hard mAP |
|---|---|---|---|---|---|---|---|---|---|---|
| AL | 200/150 | Normal | Random | – | 74.5 | 37.8 | 44.1 | 67.4 | 52.1 | 47.7 |
| | | Normal | Entropy | – | 73.6 | 48.2 | 51.9 | 71.4 | 57.9 | 53.3 |
| | | Normal | PPAL | – | 74.2 | 41.6 | 46.9 | 66.7 | 54.2 | 49.3 |
| | | Normal | CRB | – | 73.3 | 45.3 | 47.4 | 68.8 | 55.3 | 50.8 |
| | | Normal | KECOR | – | 73.2 | 46.7 | 48.2 | 69.7 | 56.0 | 51.3 |
| AL+SSL | 200/150 | Normal | Random | HSSDA | 78.8 | 54.1 | 59.9 | 77.1 | 64.3 | 59.7 |
| | | Normal | Entropy | HSSDA | 79.3 | 59.1 | 64.6 | 79.1 | 67.7 | 62.2 |
| | | Normal | PPAL | HSSDA | 80.0 | 56.1 | 66.2 | 79.7 | 67.4 | 61.8 |
| | | Normal | CRB | HSSDA | 79.0 | 58.7 | 63.9 | 78.7 | 67.2 | 62.7 |
| | | Normal | KECOR | HSSDA | 79.2 | 59.5 | 64.9 | 80.3 | 67.9 | 63.1 |
| | | Normal | CAL | HSSDA | 80.6 | 60.2 | 67.7 | 81.5 | 69.5 | 64.5 |
| $SSL_p + AL + SSL$ | 200/150 | 3DIoUMatch | Entropy | HSSDA | 78.1 | 57.3 | 64.4 | 80.1 | 66.6 | 61.5 |
| | | 3DIoUMatch | CAL | HSSDA | **80.8** | 57.1 | 65.9 | 79.9 | 67.9 | 63.1 |
| | | Joint3D | CAL | HSSDA | 78.5 | 58.9 | 70.1 | 80.7 | 69.1 | 64.1 |
| | | NAL | CAL | HSSDA | 79.8 | 59.3 | 69.9 | 81.3 | 69.6 | 64.6 |
| | | HSSDA | Entropy | HSSDA | 78.8 | 52.3 | 68.2 | 79.9 | 66.4 | 62.0 |
| | | HSSDA | CAL | HSSDA | 79.8 | 59.6 | 66.2 | 80.8 | 68.5 | 63.9 |
| $BC - SSAL$ | 200/150 | CPSP | Entropy | HSSDA | 79.5 | 57.5 | 68.0 | 80.1 | 68.3 | 62.9 |
| | | CPSP | PPAL | HSSDA | 79.9 | 55.8 | 68.1 | 80.9 | 67.9 | 62.6 |
| | | CPSP | CRB | HSSDA | 79.1 | 56.9 | 65.4 | 78.7 | 67.2 | 62.8 |
| | | CPSP | KECOR | HSSDA | 79.0 | 60.8 | 64.5 | 80.6 | 68.1 | 63.3 |
| | | CPSP | CAL | HSSDA | 79.5 | **61.2** | **70.7** | **81.8** | **70.5** | **65.1** |
| *Full* | $-/-$ | – | – | – | 84.6 | 59.6 | 72.2 | 82.7 | 72.2 | 68.5 |

## 4.2 RESULTS ON KITTI

We conduct experimental evaluation on different settings: the Active Learning (AL) framework, Semi-Supervised Active Learning (AL+SSL) framework, Pretrain-based Semi-Supervised Active Learning ($SSL_P + AL + SSL$) framework, our Bidirectional Collaborative Semi-Supervised Active Learning framework (BC-SSAL), and full-labeled (Full) results.

Among all these frameworks, they share a similar pattern. In the stage of temporary model updating, different pre-train methods are adopted like normal pre-train, 3DIoUMatch pre-train (Wang et al., 2021), Joint3D pre-train (Hwang et al., 2023), NAL pre-train (Elezi et al., 2022) and our CPSP pre-train. In the stage of unlabeled sample selection, different active learning methods are used, like Entropy, CRB (Luo et al., 2023b), KECOR (Luo et al., 2023a), PPAL (Yang et al., 2022), and our CAL. For the final model delivering stage, we leverage HSSDA (Liu et al., 2023) due to its demonstrated good performance. To improve result reliability in the limited KITTI dataset, we ran three times with different seeds and averaged the performance across them.

As shown in Table 1, our BC-SSAL framework outperforms all other approaches in average mAP, with notable improvements in challenging classes such as Pedestrian and Cyclist. When comparing pre-training methods while keeping the AL methods fixed, our CPSP pre-training consistently delivers superior performance. Notably, traditional SSL approaches negatively impact AL performance, with results declining regardless of whether 3DIoUMatch or state-of-the-art HSSDA is used. In contrast, CPSP pre-training enhances the performance of nearly all AL methods. When pre-training methods are fixed and AL methods are varied, our CAL method demonstrates superior performance across different pre-training methods, showing its effectiveness regardless of the pre-training method employed.

## 4.3 RESULTS ON WAYMO

For the Waymo dataset, to expedite the training process, we utilize CPSP in the Final Model Delivering stage. As shown in Table 2. Our BC-SSAL framework continues to demonstrate strong performance even when applied to a large amount of data, as observed in the Waymo dataset. Comparing different types of pre-training within the context of the same active learning methods, our CPSP pre-training consistently outperforms other methods in terms of average mAP, achieving an improvement of 1.2%. Notably, it significantly improves mAP for challenging classes such as Pedestrian (1.1% improvement) and Cyclist (1.9% improvement) compared to other pre-training methods using CAL. Furthermore, our proposed CAL method exhibits superior performance compared to other active learning methods. It consistently outperforms other approaches by at least 1.6% mAP within our BC-SSAL framework. More comparison with other methods with multi-rounds can be seen in the appendix D.2.

Table 2: Comparing results across different settings on the Waymo dataset. $N_1$ represents the initial box count, while $N_2$ signifies boxes selected through AL.

| Setting | $N_1/N_2$ | Pre-train | AL | SSL | Veh(L1/L2) mAP | Ped(L1/L2) mAP | Cyc(L1/L2) mAP | Avg(L1/L2) mAP | mAPH |
|---|---|---|---|---|---|---|---|---|---|
| AL | 5000/5000 | Normal | Random | – | 62.9/54.8 | 59.6/51.0 | 41.3/39.8 | 54.6/48.6 | 37.9/33.6 |
| | | Normal | Entropy | – | 61.1/53.2 | 60.6/51.9 | 50.1/48.5 | 57.3/51.2 | 40.4/35.9 |
| | | Normal | CRB | – | 62.7/54.4 | 56.6/48.4 | 54.6/52.7 | 57.9/51.8 | 38.8/34.4 |
| | | Normal | KECOR | – | 61.8/53.5 | 57.1/49.0 | 52.1/50.2 | 57.0/50.8 | 39.1/34.5 |
| AL+SSL | 5000/5000 | Normal | Random | CPSP | 63.1/54.8 | 59.4/50.0 | 46.1/45.3 | 56.2/50.0 | 40.3/36.8 |
| | | Normal | Entropy | CPSP | 61.5/53.5 | 60.0/51.5 | 54.9/53.0 | 58.8/52.7 | 43.4/38.9 |
| | | Normal | CRB | CPSP | 63.2/54.6 | 57.2/48.9 | 56.9/54.2 | 59.1/52.6 | 42.1/37.7 |
| | | Normal | KECOR | CPSP | 62.5/53.8 | 57.5/49.7 | 56.1/53.6 | 58.7/52.4 | 42.3/38.1 |
| | | Normal | CAL | CPSP | 62.2/54.3 | 61.7/53.0 | 55.4/53.5 | 59.8/53.6 | 45.1/40.6 |
| $SSL_p + AL + SSL$ | 5000/5000 | 3DIoUMatch | CAL | CPSP | 63.1/54.7 | 60.9/52.7 | 53.3/51.6 | 59.1/53.0 | 43.9/39.2 |
| | | Joint3D | CAL | CPSP | 61.5/53.9 | 59.7/51.3 | 53.2/51.8 | 58.1/52.3 | 41.8/37.2 |
| | | NAL | CAL | CPSP | 62.6/54.3 | 60.1/51.7 | 55.1/53.1 | 59.3/53.1 | 42.0/38.1 |
| $BC-SSAL$ | 5000/5000 | CPSP | Entropy | CPSP | **64.2/56.2** | 60.3/50.8 | 53.4/51.5 | 59.3/52.8 | 44.1/40.7 |
| | | CPSP | CRB | CPSP | 63.9/55.0 | 59.1/49.1 | 53.8/51.8 | 58.9/52.0 | 43.0/39.0 |
| | | CPSP | KECOR | CPSP | 63.0/54.5 | 60.2/50.6 | 54.3/52.4 | 59.2/52.5 | 44.6/41.1 |
| | | CPSP | CAL | CPSP | 62.8/54.8 | **62.8/54.1** | **57.3/55.3** | **61.0/54.7** | **46.5/42.4** |
| *Full* | –/– | – | – | – | 75.4/67.4 | 72.0/63.7 | 65.9/63.4 | 71.1/64.8 | 66.7/60.9 |

Table 3: Ablation study of different components in CPSP.

| Pre-train | Stage | 3D Detection | | | mAP |
|---|---|---|---|---|---|
| | | Car | Ped. | Cyc. | |
| Normal | TMU | 71.8 | 30.0 | 14.9 | 38.9 |
| Normal | FMD | **80.6** | 60.2 | 67.7 | 69.5 |
| CPSP w/o iter | TMU | 77.7 | 42.7 | 27.4 | 49.3 |
| CPSP w/o iter | FMD | 79.3 | 61.1 | 70.1 | 70.2 |
| CPSP w/ iter | TMU | 77.8 | 44.3 | 36.1 | 52.7 |
| CPSP w/ iter | FMD | 79.5 | **61.2** | **70.7** | **70.5** |

Table 4: Ablation study of different components in CAL. mAP is calculated under the moderate difficulty level.

| CAL | | | 3D Detection | | | mAP |
|---|---|---|---|---|---|---|
| $E^2\_Unc$ | $CBS$ | $B\_Div$ | Car | Ped. | Cyc. | |
| - | - | - | 78.8 | 54.1 | 59.9 | 64.3 |
| ✓ | - | - | **80.9** | 58.7 | 67.6 | 69.1 |
| ✓ | ✓ | - | 79.6 | 61.0 | 70.2 | 70.3 |
| ✓ | ✓ | ✓ | 79.5 | **61.2** | **70.7** | **70.5** |

## 4.4 ANALYSIS

In this section, we present a series of ablation studies to analyze the effect of our proposed strategies in BC-SSAL.

### 4.4.1 ABLATION STUDY OF CONFIDENT OBJECT EXTRACTION (CPSP)

Table 3 demonstrates the results of our ablation study. It indicates that both the introduction of the pseudo scene and the iterative refinement mechanism contribute positively to the model's performance. During the pre-training stage in TMU (Temporal Model Updating), CPSP achieves substantial improvements in mAP scores for all classes, with total mAP enhancements of 9.4% (without iteration) and 13.8% (with iteration). Notably, these improvements are particularly pronounced in challenging classes such as Pedestrian and Cyclist, where we achieve gains exceeding 10% in mAP. Besides, these enhancements carry over to the final model delivering stage, resulting in higher overall mAP scores compared to normal pre-training. The most significant gains are observed in the challenging classes, with an improvement of 1% in mAP for pedestrians and 3% for cyclists.

### 4.4.2 ABLATION STUDY OF COLLABORATIVE ACTIVE LEARNING (CAL)

The ablation study, as depicted in Table 4, emphasizes the importance of the uncertainty measure, class balance methods, and diversity methods in CAL for achieving improved performance. The $E^2\_Unc$ plays a crucial role in active learning by selecting informative samples. This selection process enables the model to focus on challenging instances, leading to an overall mAP improvement of 4.8%. CBS contributes significantly to addressing class imbalances, particularly in hard classes like Pedestrian(2.3% mAP improvement) and Cyclist(2.6 % mAP improvement), resulting in enhanced performance in these challenging scenarios. Additionally, $B\_Div$ helps reduce redundancy in the selected samples, enabling the model to capture a broader range of object variations and further improve its detection capabilities. By incorporating these CAL components, the overall semi-supervised active learning framework becomes more effective, leading to better performance in 3D object detection.

### 4.4.3 ANALYSIS ABOUT DIFFERENT PRE-TRAINING METHODS.

To evaluate how well our CPSP pre-trained model aligns with uncertainty-based active learning methods for object detection, we focus on two key aspects: Calibration and Detection Performance.

Calibration (Guo et al., 2017) refers to how accurately the model's confidence scores reflect the correctness of its predictions. A well-calibrated model is crucial for active learning, as it helps in

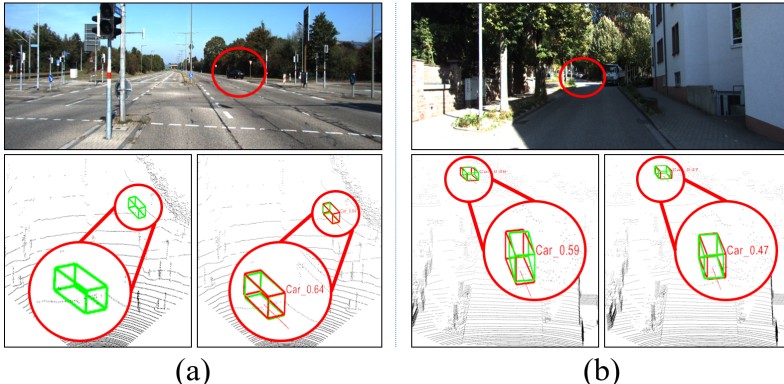

Figure 4: Qualitative results of selected samples. Green boxes represent GT boxes, while the red boxes denote the predicted boxes. We visualize two scenes, one located on the left(a) and the other on the right(b). Each scene is presented with three images: the top image shows the corresponding 2D image, the bottom-left image displays the predicted results from the normal pre-trained model, and the bottom-right image shows the predicted results from the CPSP pre-trained model.

selecting the most informative samples. We use D-ECE (Kuppers et al., 2020) to measure the calibration quality. Detection Performance, on the other hand, measures the overall detection ability of the model and is quantified by mAP. As noted in appendix E.3, the KITTI dataset includes many 'DontCare' labels, making it challenging to accurately calculate D-ECE scores. Therefore, we conduct our analysis using the Waymo training set. As shown in Table 5, our CPSP model achieves strong performance in both D-ECE and mAP, better supporting the active learning process. In contrast, other methods perform poorly in either D-ECE or mAP, making them less effective for active learning. Please see appendix E.1 for more analysis.

Table 5: D-ECE scores, mAP(LEVEL_1) for different pre-train methods on Waymo training set.

| Pre-train | D-ECE ↓ | | | mAP(%) | | | |
|---|---|---|---|---|---|---|---|
| | Veh | Ped. | Cyc. | Veh | Ped. | Cyc. | Avg |
| Normal | 0.11 | 0.10 | 0.25 | 58.7 | 53.9 | 31.9 | 48.2 |
| 3DIoUMatch (Wang et al., 2021) | 0.50 | 0.13 | 0.29 | 57.4 | 45.4 | 33.0 | 45.3 |
| Joint3D (Hwang et al., 2023) | 0.30 | 0.36 | 0.48 | 59.3 | 50.0 | 38.1 | 49.1 |
| NAL (Elezi et al., 2022) | 0.28 | 0.26 | 0.42 | 59.5 | 48.2 | 39.0 | 49.0 |
| CPSP | **0.09** | **0.08** | **0.15** | **60.4** | **55.1** | **35.2** | **50.2** |

### 4.4.4 QUALITATIVE RESULTS

We present visualizations of selected samples in Fig. 4. In Fig. 4(a), we observe that our CPSP pre-trained model is capable of detecting hard objects that are missed by a model trained with normal pre-training. This highlights the effectiveness of our CPSP approach in discovering challenging objects. In Fig. 4(b), we showcase how our CPSP pre-trained model retains uncertainty for real unconfident boxes. This ability to maintain uncertainty is crucial for effective active learning, enabling the model to focus on challenging examples and improve its performance.

## 5 CONCLUSION

In this paper, we propose a **B**idirectional **C**ollaborative **S**emi-**S**upervised **A**ctive **L**earning framework, dubbed as BC-SSAL, which consists of Collaborative PseudoScene Pre-training (CPSP) and Collaborative Active Learning (CAL), effectively addressing the conflicts between semi-supervised learning and active learning. CPSP utilizes pseudo scenes with confident boxes for model pre-training, while CAL maximizes the benefits of the CPSP pre-trained model to select superior samples. Experimental results on KITTI and Waymo datasets demonstrate that our approach achieves state-of-the-art performance, offering a promising solution for improving 3D object detection through effective integration of semi-supervised and active learning.

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

In the **Appendix**, we provide further details about our BC-SSAL method in Sec. B. Additionally, we include more implementation specifics in Sec. C and extended experimental results in Sec. D. Further analysis and visualizations are presented in Sec. E, offering a more comprehensive evaluation of BC-SSAL's performance. For ease of reference, a list of abbreviations used throughout the paper is provided in Sec. A.

## A    LIST OF TITLE WORD ABBREVIATIONS

| Abbreviation | Full Title |
|---|---|
| AL | Active Learning |
| SSL | Semi-supervised Learning |
| SSAL | Semi-supervised Active Learning |
| TMU | Temporary Model Updating Stage |
| USS | Unlabeled Sample Selecting Stage |
| FMD | Final Model Delivering Stage |
| CPSP | Collaborative Pseudo-Scene Pre-training |
| CAL | Collaborative Active Learning |
| $E^2\_Unc$ | Ensemble-based Entropy Uncertainty |
| $B\_Div$ | Box-level Diversity |
| CBS | Class Balance Sampling |

## B    MORE METHOD DETAILS FOR BC-SSAL

### B.1    COLLABORATIVE PSEUDOSCENE PRE-TRAINING

In this section, we delve into the details of extracting confident objects during the Collaborative PseudoScene Pre-training.

**More details about Confident Object Extraction**    To filter confident objects, previous SSL methods often use fixed thresholds or top-k selections. However, we observed that score distributions vary significantly across different classes and models, making it challenging to determine an appropriate class-specific threshold. Additionally, the performance of different models can fluctuate, complicating the selection of a consistent top-k value—if $k$ is too small, the number of extracted objects may be insufficient, while a larger $k$ introduces more noise. To address this, we employ clustering methods (Liu et al., 2023), such as KMeans, on uncertainty scores to select confident objects. Clustering allows objects within the same group to share similar patterns, and by choosing a relatively higher number of centers, we improve the reliability of confident object selection.

Besides, to continuously improve the box bank, we implement an iterative refinement mechanism that selectively incorporates newly extracted confident objects. Let $O_{\text{new}}$ represent the set of newly extracted objects and $O_{\text{bank}}$ represent the set of objects already stored in the box bank. For each new object $o_{\text{new}} \in O_{\text{new}}$, if it overlaps with an existing object $o_{\text{bank}} \in O_{\text{bank}}$, we compare their uncertainty scores, denoted as $U(o_{\text{new}})$ and $U(o_{\text{bank}})$ respectively. The object with the lower uncertainty score is retained:

$$o_{\text{retain}} = \begin{cases} o_{\text{new}}, & \text{if } U(o_{\text{new}}) < U(o_{\text{bank}}) \\ o_{\text{bank}}, & \text{otherwise} \end{cases} \tag{3}$$

This ensures that only higher-confidence objects are kept in the box bank, minimizing the risk of introducing noisy or uncertain objects into the training data.

Even though the retained objects have lower uncertainty scores, errors can still occur during the uncertainty estimation, potentially leading to the inclusion of mislabeled or erroneous objects. To address this issue, we introduce a deletion mechanism. After each training iteration, we extract new confident boxes and compare them to existing boxes in the bank. The deletion mechanism checks for discrepancies between newly extracted objects $o_{\text{new}}$ and existing objects $o_{\text{bank}}$. If the overlap between two objects exceeds a certain threshold $\tau_{\text{overlap}}$, defined as the Intersection over Union (IoU):

$$\text{IoU}(o_{\text{new}}, o_{\text{bank}}) > \tau_{\text{overlap}}, \tag{4}$$

we retain the object with the lower uncertainty score and delete the other. Additionally, mislabeled or erroneous objects are identified and removed by evaluating their performance in subsequent training iterations.

This iterative process of adding, comparing, and refining the box bank improves its overall quality, ensuring that the uncertainty estimation becomes more reliable over time. Mathematically, this can be formalized as:

$$O_{\text{bank}}^{t+1} = \left(O_{\text{bank}}^{t} \setminus O_{\text{remove}}\right) \cup O_{\text{new}}, \tag{5}$$

where $O_{\text{bank}}^{t}$ is the set of objects in the box bank at iteration $t$, and $O_{\text{remove}}$ is the set of objects identified for deletion based on the comparison with $O_{\text{new}}$. The iterative refinement ensures that $O_{\text{bank}}$ evolves to contain higher-quality pseudo-labels for training.

### B.2 COLLABORATIVE ACTIVE LEARNING

In this part, we provide more details about the Class Balance Sampling. We also present the completed pseudo-code for our active learning process, as shown in Algorithm 1.

**More details about Class Balance Sampling** Outdoor LiDAR scenes often present challenges due to the class imbalance, where some classes, such as cyclists, are rare compared to more frequent classes like cars. These rare classes are difficult to sample and annotate effectively. To address this, we propose a class balance sampling algorithm that adjusts the number of samples for each class based on its difficulty and co-occurrence patterns.

**Class Weight Calculation:** For each class $c$, we compute the average uncertainty $u_c$ of all predicted boxes from the unlabeled data. The uncertainty of a class reflects the model's performance on that class, with higher uncertainty suggesting more difficulty in correctly predicting objects of that class.

Using this uncertainty, we assign a class weight $w_c$ that prioritizes classes with higher uncertainty, ensuring that more challenging classes receive more attention during the sampling process. The class weight $w_c$ is computed as follows:

$$w_c = \frac{\sqrt{1/u_c}}{\sum_c \sqrt{1/u_c}}, \tag{6}$$

where the reciprocal square root of the uncertainty is taken to ensure that more uncertain classes receive higher weights. This weight is then normalized across all classes. The goal of this weighting is to prioritize the selection of challenging classes while maintaining a balance across the dataset.

**Class Co-occurrence Patterns:** In outdoor scenes, multiple object classes often appear together. Therefore, simply assigning weights based on uncertainty may not fully address the issue of class imbalance, especially when considering the co-occurrence of objects from different classes in the same scene.

To handle this, we record the co-occurrence patterns between classes. For example, if a scene contains a car, we examine the average frequency of other classes (e.g., cyclists) appearing alongside it. This co-occurrence data is stored in a co-occurrence matrix $A$, where each entry $A_{ij}$ represents the likelihood of class $j$ appearing in a scene when class $i$ is present.

Let's denote $A$ as a $C \times C$ matrix, where $C$ is the number of classes. Each row corresponds to a specific class and indicates the average appearance of other classes when that class is present. The diagonal elements of $A$ are normalized to 1 to indicate the expected number of objects for that class when it is present.

Using this co-occurrence matrix, we compute the desired upper limits $U$ for the number of objects to sample from each class. We want these upper limits to reflect the desired sampling ratio while considering the class co-occurrence patterns and uncertainty-based weights. This can be expressed by the following equation:

$$AU = W \times B, \tag{7}$$

---

**Algorithm 1** Collaborative Active Learning Algorithm

---

**Input**:

- Labeled data $D_l = \{S_{li}\}_{i=1}^{N_l}$
- Unlabeled data $D_u = \{S_{ui}\}_{i=1}^{N_u}$
- Budget $b$ for selecting new samples
- Class set $C$
- Similarity threshold $T_{sim}$
- Class upper limits $U(c)$ for each class $c \in C$
- Weight adjustment factor $S(c)$ for each class $c \in C$

**Output**: Selected data $\Delta D$ for labeling
**Initialize**: $\Delta D \leftarrow \varnothing$ (selected set of data to be labeled)

1: Calculate uncertainty of each sample in $D_u$ using $E^2\_Unc$: $\{E_i\}_{i=1}^{N_u}$.
2: Sort $D_u$ in descending order based on uncertainty values $\{E_i\}$.
3: Initialize $idx \leftarrow 0$ and $Box(c) \leftarrow 0$ (the count of boxes for each class $c \in C$).
4: **while** $Num_{box}(\Delta D) < b$ **do**
5:     Compute similarity $S_{idx}$ between $D_u(idx)$ and the already selected data $\Delta D$ using $B\_Div$.
6:     **if** $S_{idx} < T_{sim}$ **then**
7:       Add $D_u(idx)$ to $\Delta D$ and remove it from $D_u$.
8:       **if** there exists class $c \in C$ such that $Box(c) < U(c)$ and $Box(c)+Num_{box}(D_u(idx,c)) \geq U(c)$ **then**
9:         Recalculate uncertainties $\{E_i\}_{i=1}^{N_u}$ considering weight adjustment factor $S(c)$.
10:        Resort $D_u$ based on the updated uncertainties.
11:        Set $idx \leftarrow 0$.
12:       **else**
13:        Set $idx \leftarrow idx + 1$.
14:       **end if**
15:       Update $Box(c)$ for each class based on newly selected data.
16:     **end if**
17: **end while**

---

where: $A$ is the co-occurrence matrix, $U$ is the vector of upper limits for each class, $W$ is the vector of class weights computed using Equation 6, and $B$ is the total number of boxes to be selected.

By solving this linear system, we can determine a balanced sampling strategy that takes both co-occurrence patterns and class uncertainty into account.

**Handling Small or Negative Upper Limits:** One challenge we might encounter when solving for the upper limits $U$ is that some values could be unrealistically small or even negative, which would not be appropriate for a balanced sampling strategy. To address this, we impose a constraint that ensures each upper limit $U_c$ is greater than or equal to a minimum threshold, such as $B/10$, where $B$ is the total number of samples to be selected.

Once the upper limits are determined, we apply these limits during the sampling process. If the number of selected samples for a particular class exceeds its upper limit, the uncertainty scores for that class are decreased accordingly, ensuring that the algorithm maintains a balanced representation of all classes.

This approach ensures that rare or difficult classes are not underrepresented, while also preventing the model from over-sampling classes that are easier to detect.

**The Pipeline of Collaborative Active Learning** The pipeline of collaborative active learning is demonstrated in Algorithm 1. First, we calculate the uncertainty of all unlabeled data and sort them in descending order. Next, we adopt a greedy approach to select the data with the highest uncertainty. Throughout this process, we discard samples that are similar to the ones already selected to ensure

diversity. Moreover, we incorporate class balance considerations by adjusting the uncertainty of a class when it reaches its upper limit.

By following this pipeline, we systematically evaluate uncertainty, prioritize the most unconfident samples, promote diversity by excluding similar data points, and account for class balance by adjusting uncertainty based on upper limits.

**The Pipeline of Collaborative Active Learning** The pipeline for Collaborative Active Learning (CAL) is outlined in Algorithm 1. The algorithm operates in the following steps:

1. **Uncertainty Calculation:** For each sample in the unlabeled dataset, we compute the uncertainty score using our Ensemble-based Entropy Uncertainty ($E^2\_Unc$) method. This uncertainty quantifies how confident the model is about each prediction.

2. **Sorting by Uncertainty:** The unlabeled data is sorted in descending order of uncertainty, so that the most uncertain samples are prioritized for selection.

3. **Diversity Enforcement:** To maintain diversity among the selected samples, we calculate the similarity between each new candidate and the previously selected samples using Box-level Diversity ($B\_Div$). If the similarity exceeds a predefined threshold $T_{sim}$, the candidate sample is discarded to avoid redundancy.

4. **Class Balance Adjustment:** During the selection process, we also enforce class balance by setting upper limits on the number of objects that can be selected from each class. If a class exceeds its upper limit $U(c)$, we adjust the uncertainty scores for that class by applying a weight factor $S(c)$. This recalibration ensures that we avoid over-sampling from any single class.

5. **Greedy Selection:** The algorithm proceeds in a greedy manner, selecting the most uncertain and diverse samples until the budget $b$ is reached.

By following this pipeline, our method efficiently selects the most informative samples for labeling, ensuring both diversity and class balance while focusing on high-uncertainty data points. This balanced approach improves the effectiveness of active learning, reducing redundancy and ensuring that the model is trained on representative and diverse data.

## C  MORE IMPLEMENTATION DETAILS

In the Confident Object Extraction module, we use K-Means to do clustering and set the number of centers as 20 for the KITTI dataset and 50 for the Waymo dataset. The similarity threshold ($T_{sim}$) in the $B\_Div$ module is set to 0.9. It is worth noting that due to the limited amount of data utilized, achieving model convergence can be challenging. To address this, we extend the training iterations, allowing the model to train for a longer duration. In order to achieve this, we repeat the data length 5 times for the KITTI dataset and 15 times for the Waymo dataset. Other basic settings like learning rate, optimizer, and scheduler are following (Shi et al., 2020; Team, 2020).

Furthermore, when the normal pre-trained model undergoes the Collaborative Active Learning (CAL) module, we utilize the upper limits calculated by the CPSP pre-trained model to ensure a fair comparison. By incorporating these strategies, we can effectively evaluate the performance of the normal pre-trained model in comparison to the CPSP pre-trained model. This ensures a reliable and comprehensive assessment of the effectiveness of our proposed method.

## D  ADDITIONAL EXPERIMENT

### D.1  MORE EXPERIMENTS ABOUT DIFFERENT SEMI-SUPERVISED METHODS ON KITTI

We replace the SSL methods in Tab.1 with our Collaborative PseudoScene Pre-training (CPSP) approach in the Final Model Delivering Stage (FMD) to examine how well our method performs when compared with different pre-train methods. As presented in Tab. F, the results demonstrate that our CPSP method not only achieves the best performance but also generates a more significant performance gap in comparison to other pre-train methods. The improvement is particularly noticeable

across challenging classes such as pedestrians and cyclists. This highlights the robustness of CPSP in handling diverse object detection tasks.

Table F: Comparison of results for various methods on the KITTI dataset, with all frameworks using the same amount of labeled data. $N_1$ denotes the initial number of boxes, and $N_2$ represents boxes selected during active learning.

| Pre-train | AL | SSL | $N_1/N_2$ | Car_mod mAP | Ped_mod mAP | Cyc_mod mAP | Avg_easy mAP | Avg_mod mAP | Avg_hard mAP |
|---|---|---|---|---|---|---|---|---|---|
| Normal | CAL | CPSP | 200/150 | 76.8 | 54.7 | 67.3 | 79.9 | 66.3 | 61.2 |
| Joint3D | CAL | CPSP |  | 77.6 | 55.1 | 63.4 | 77.2 | 65.3 | 60.8 |
| NAL | CAL | CPSP | 200/150 | 77.3 | 60.8 | 62.6 | 78.7 | 66.9 | 61.2 |
| HSSDA | CAL | CPSP |  | 77.8 | 60.7 | 66.0 | 80.2 | 68.2 | 62.8 |
| CPSP | CAL | CPSP | 200/150 | **79.3** | **62.2** | **68.3** | **82.5** | **69.9** | **64.5** |

## D.2 Multiple Rounds of Experiments

We conducted experiments using both single-round and multi-round approaches to assess their potential for improving performance. Specifically, we performed experiments on the Waymo dataset, utilizing a total of 20,000 annotated boxes to compare the outcomes of single-round versus three-round approaches. As shown in Table G, increasing the number of rounds while maintaining the same annotation budget results in a 0.9% improvement in mAP. We attribute this enhancement to the model's improved ability to select better data over multiple rounds, underscoring the positive impact of utilizing multiple rounds in our approach.

In addition, to facilitate a comprehensive comparison of our methods with existing approaches, we employ active learning over multiple rounds. In each round, we annotate 5000 boxes, conducting a total of 5 rounds, which culminate in 30,000 annotated boxes. We compare the following methods in our evaluation: Joint3D (Hwang et al., 2023), NAL (Elezi et al., 2022), 3DIoUMatch (Wang et al., 2021) combined with our CAL, Normal pre-training combined with CAL, and our CPSP pre-training combined with CAL. For a fair comparison, all methods utilize CPSP in the Unlabeled Sample Selecting Stage. As depicted in Fig. E, our method (CPSP + CAL) consistently outperforms all other approaches across every round of active learning, demonstrating substantial performance gains even in the final evaluation round. The sustained advantage of our method highlights its effectiveness in selecting and leveraging superior data throughout the active learning process.

Table G: Results of single-round and three-round on Waymo.

| Setting | Rounds | Numbers | Veh(L1/L2) | Ped(L1/L2) | Cyc(L1/L2) | Avg(L1/L2) |
|---|---|---|---|---|---|---|
| CPSP + CAL | 1 | 5000+15000 | 65.3/57.1 | 65.0/56.2 | 58.1/55.9 | 62.8/56.3 |
| CPSP + CAL | 3 | 5000+3×5000 | **65.8/57.6** | **66.0/57.2** | **59.2/56.8** | **63.7/57.3** |

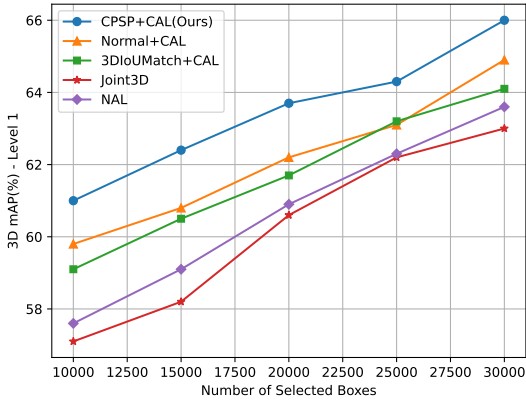

Figure E: Performance(mAP) of different methods in multiple rounds on Waymo Datasets.

# E  ADDITIONAL ANALYSIS

## E.1  ANALYSIS OF DIFFERENT PRE-TRAINING METHODS

To gain a deeper understanding of why our CPSP pre-training method outperforms other pre-training methods, we further analyze the model's calibration beyond the D-ECE score. Specifically, we divide the confidence scores into four ranges: 0-0.3, 0.3-0.5, 0.5-0.8, and 0.8-1. The scores in the 0.3-0.8 range represent more uncertain predictions, which are more likely to be selected by active learning, while the 0.8-1 range corresponds to predictions that are more likely to represent real objects.

As shown in Fig. F, our CPSP method demonstrates superior calibration, particularly in the 0.3-0.8 range, which is critical for active learning tasks. The accuracy of predictions within this range is significantly higher compared to other pre-training methods, indicating that CPSP better handles uncertain predictions and reduces overconfidence. Moreover, CPSP consistently outperforms across all three object classes, maintaining both higher precision in high-confidence predictions (0.8-1 range) and better accuracy for uncertain predictions (0.3-0.8 range). This balanced calibration makes CPSP especially effective for active learning, where selecting informative samples from uncertain predictions is crucial for optimizing model performance with limited data.

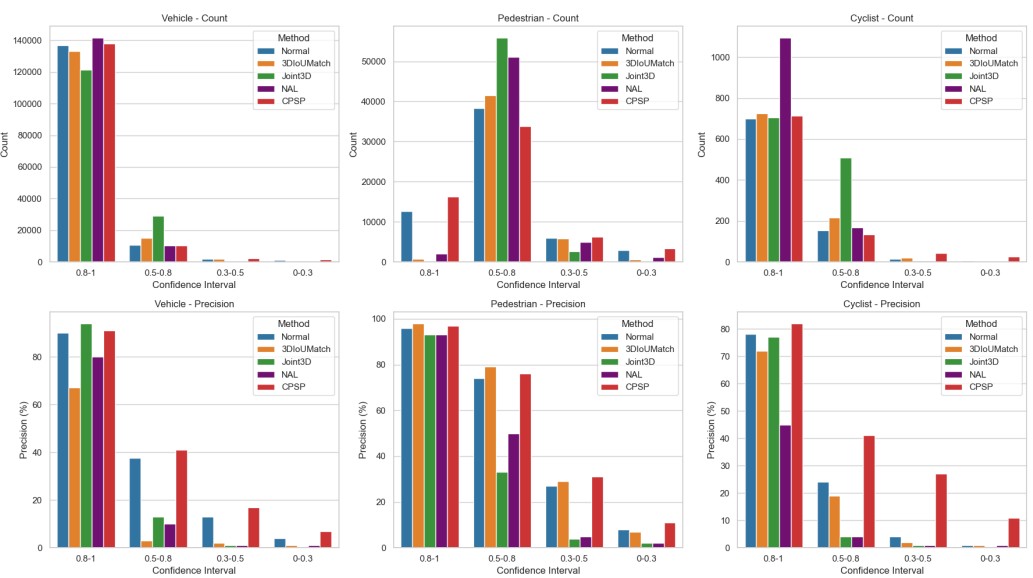

Figure F: Histogram illustrating the distribution of Precision and Count across different pre-training methods for three object classes. The figure highlights the comparative performance of each method in terms of prediction accuracy and the number of predictions within specific confidence intervals.

## E.2  NEGATIVE EFFECTS OF TRAINING ON UNCONFIDENT OBJECTS

Confident objects provide crucial information with minimal noise, while unconfident ones introduce numerous incorrect pseudo-labels that mislead the model. We utilize two types of pre-training in our approach CPSP: UNC (Unconfident) and CON (Confident). In UNC pre-training, objects with a low number of clustering centers (2) are filtered, allowing us to include more unconfident objects in the model training process. In CON pre-training, we train confident objects using the same methods but with a higher number of clustering centers (20). This approach allows us to focus on objects that exhibit a higher level of certainty and reliability during the training process. Fig. G depicts the comparison between UNC and CON on the KITTI dataset. UNC is adversely affected by false positives, degrading uncertainty assessment, and yielding inferior results (69.2%). In contrast, CON achieves a higher score of 70.5%.

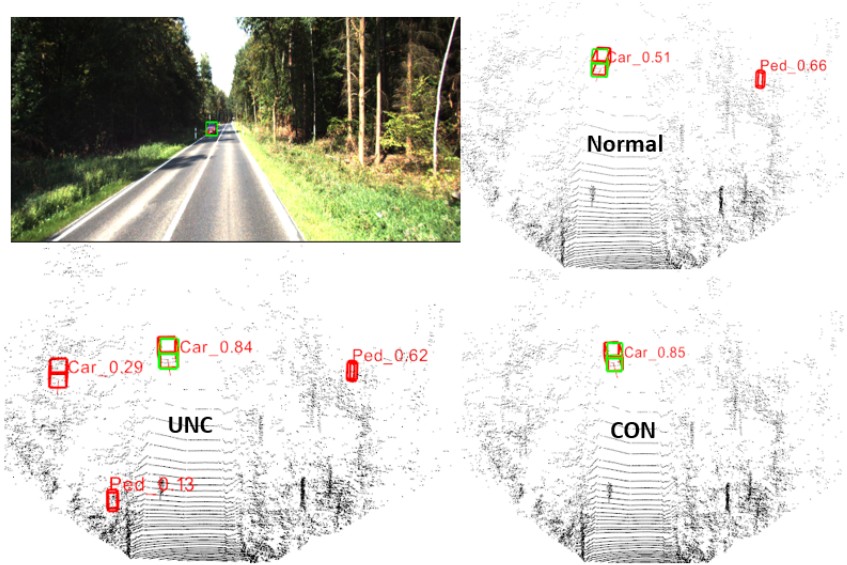

Figure G: In Normal, an unconfident car and a false positive pedestrian (tree) are detected. UNC is impacted by noise from unconfident labels, leading to more false positives. In contrast, CON learns the car effectively and eliminates the false positives.

### E.3 VISUALIZATION OF SELECTED SAMPLES WITHOUT DROPPING "DONTCARE" CASES

As depicted in Fig. H, the visualization showcases selected examples that include "DontCare" areas. These "DontCare" areas often contain numerous challenging objects with high uncertainty. Consequently, these objects contribute significantly to the uncertainty measure of the frames. However, since they are unlabeled, their presence can potentially hinder the performance of active learning methods. In the context of evaluating active learning methods, it is necessary to exclude "DontCare" cases. To accomplish this, we employ a specific procedure. First, we project our predicted 3D boxes onto 2D images since "DontCare" areas only have 2D annotations in the KITTI dataset. Next, we check whether the centers of the projected 2D boxes are located within the "DontCare" areas. If we find that more than two predicted boxes are situated within these "DontCare" areas, we exclude the respective frame from our selection.

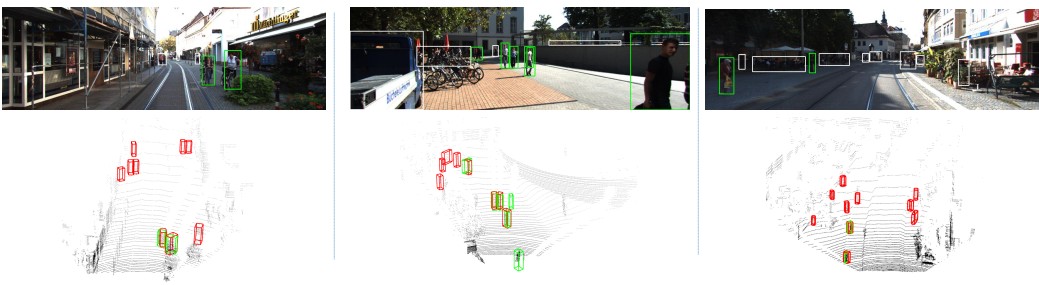

Figure H: This figure showcases the visualization of selected samples without dropping the "Dont-Care" cases. It displays the ground-truth (GT) boxes in green, the predicted boxes in red, and the "DontCare" areas in white. Each scene is presented through a 2D image and a point cloud representation. In the 2D images, both the GT boxes and the "DontCare" areas are visualized, while in the point cloud scenes, both the GT and predicted boxes are visualized.

### E.4    VISUALIZATION OF SELECTED SAMPLES

As shown in Fig. I, it provides a visual representation of the selected samples that highlight challenging instances across various object classes. The chosen samples encompass a diverse range of scenarios and objects, capturing challenging cases that require accurate detection and localization. The visualization of these challenging samples demonstrates the effectiveness of our active learning strategy in identifying and prioritizing hard examples.

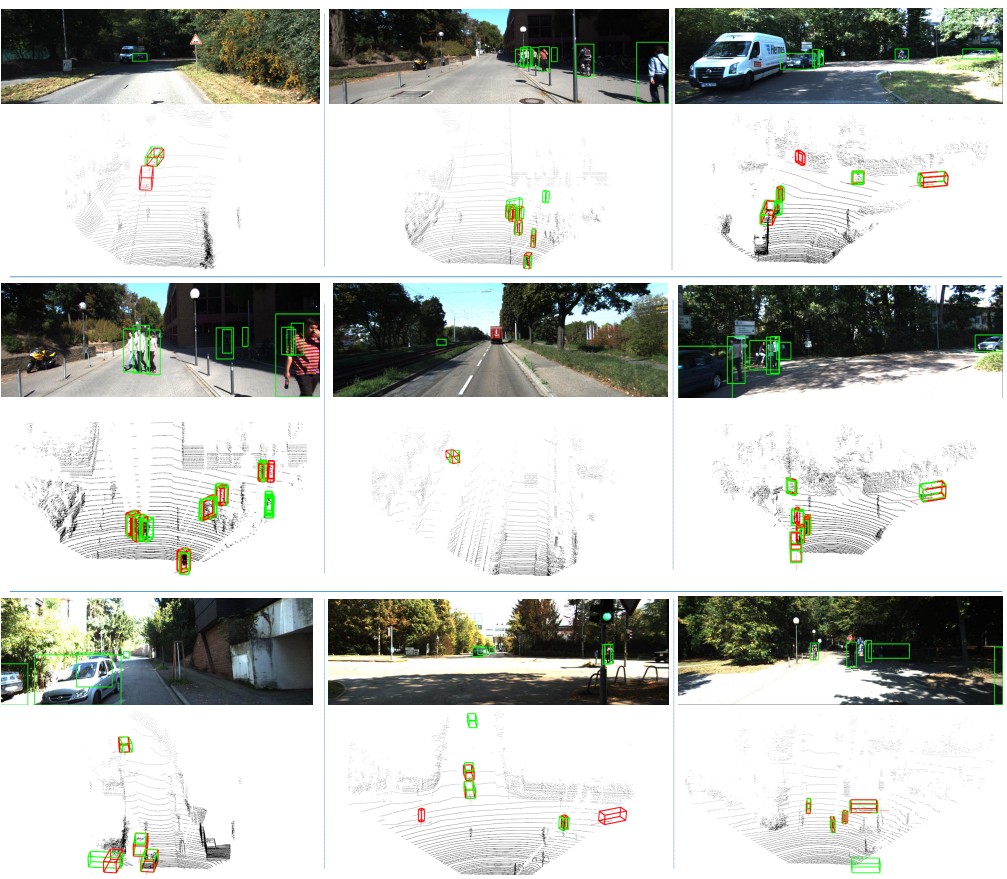

Figure I: This figure displays the visualization of selected samples, showcasing the ground-truth (GT) boxes in green and the predicted boxes in red. Each scene is represented by both a 2D image and a point cloud. In the 2D images, only the GT boxes are visualized, while in the point clouds, both the GT and predicted boxes are visualized.

