# OpenReview forum: "Unleashing the Potential of Unlabeled Data: Bidirectional Collaborative Semi-Supervised Active Learning for 3D Object Detection"
_ICLR.cc/2025/Conference — Submitted to ICLR 2025_

### Official Review · Reviewer_Dsaa · 2024-11-02

**Soundness:** 2
**Presentation:** 2
**Contribution:** 2
**Rating:** 5
**Confidence:** 5

**Summary:**

This paper proposes an active learning and semi-supervised learning collaborative label-efficient 3D object detection method. By mining useful information from unlabeled data to enhance the performance of 3D detectors. The proposed method is experimentally validated on two widely used datasets.

**Strengths:**

1. The paper presents an ideal for 3D object detection that collaborates semi-supervised learning and active learning bidirectionally, which is novel and interesting.
2. The design of the active learning scheme is reasonable, and the experiments have successfully validated the performance improvement brought by this module to the entire method.

**Weaknesses:**

1. The proposed method has limited innovation; the CPSP module is highly similar to the 'Reliable Background Mining Module' presented in the existing SS3D[1]. The Reliable Background Mining Module employs a pre-trained detector to process unlabeled scenes, extracting foreground instances to construct a bank, and then uses gt-sampling data augmentation to generate new training data. The CPSP module follows the same procedure without any difference.

2. The motivation behind this paper is unclear, as there is no data proving the conflict between SSL and AL. It is suggested to provide experimental results of a naive combination of SSL  and AL  for 3D object detection, demonstrating that directly combining the two methods is unreliable. Additionally, the method does not offer a special design based on the proposed concept of 'bidirectional collaboration'.

3. The figures in the paper are sketchy and not aesthetically pleasing.

[1] Liu. et.al. SS3D: Sparsely-Supervised 3D Object Detection from Point Cloud. CVPR2022

**Questions:**

see Weaknesses*.

---

> ### Author Response · Authors · 2024-11-19
> **Response to Reviewer Dsaa**
>
> Dear Reviewer Dsaa,
>
> We sincerely appreciate your review with thoughtful comments. We have carefully considered each of your questions and provide detailed responses below to clarify any misunderstandings. Please let us know if you have any further questions or concerns.
>
> **Q1: The difference between our CPSP and SS3D[1].**
>
> A1:  While our CPSP shares some conceptual similarities with SS3D, it differs fundamentally in both design and purpose. The key distinction between our CPSP and existing SS3D lies in **how pseudo-objects are selected for training**. SS3D is designed for a sparse-supervised setting, focusing on extracting as many unlabeled objects as possible, whereas CPSP is specifically tailored to support the AL process. This fundamental difference leads SS3D to overlook two critical aspects that are central to CPSP: the **box bank deletion mechanism** and the **use of backgrounds likely to be false positives (FPs) for training**. Let’s delve into these two factors to illustrate their importance.
>
> - **Box Bank Deletion Mechanism:**
>
>   No matter how confident or refined the filtering process is, pseudo-labels in the box bank inevitably contain errors and noise. Without a mechanism to delete these noisy labels, they propagate throughout training, resulting in confirmation bias that adversely affects uncertainty measurement. CPSP incorporates a robust deletion mechanism to address this issue, ensuring cleaner pseudo-labels and better AL outcomes.
>
> - **Use of Backgrounds Likely to Be FPs:**
>
>   CPSP utilizes background regions that are likely to be FPs, providing **negative signals** to the model. These negative signals help the model avoid overconfidence in incorrect predictions, thereby improving the reliability of uncertainty measures.
>
> To further demonstrate these differences, we conducted experiments by directly applying SS3D  within our framework. The results, presented below, compare the mAP and D-ECE (calibration, where lower is better) metrics for various configurations:
>
> Table I: mAP comparison across settings on Waymo
>
> | Settings   | Veh           | Ped           | Cyc           | Avg           |
> | ---------- | ------------- | ------------- | ------------- | ------------- |
> | Normal+CAL | 62.2/54.3     | 61.7/53.0     | 55.4/53.5     | 59.8/53.6     |
> | SS3D+CAL   | 62.7/54.8     | 60.2/51.9     | 54.2/52.1     | 59.1/52.9     |
> | CPSP+CAL   | **62.8/54.8** | **62.8/54.1** | **57.3/55.3** | **61.0/54.7** |
>
> Table II: D-ECE comparison across different pre-training methods on Waymo
>
> | Pre-train | Veh      | Pe       | Cyc      |
> | --------- | -------- | -------- | -------- |
> | Normal    | 0.11     | 0.10     | 0.25     |
> | SS3D      | 0.11     | 0.16     | 0.38     |
> | CPSP      | **0.09** | **0.08** | **0.15** |
>
> The results indicate that SS3D negatively impacts calibration (Table II), particularly for harder classes (pedestrians and cyclists), thereby limiting its effectiveness in supporting the AL process (Table I). CPSP, on the other hand, demonstrates significant improvements in both performance and calibration, validating the necessity of its design choices.
>
> **Q2:  The motivation of the conflict between SSL and AL.**
>
> A2:  Actually, **there are many evidences** in our paper showing the conflict between SSL and AL. In Tables 1 and 2, various SSL pre-training methods combined with AL methods often perform worse than their counterparts (Normal + AL). We also provide a detailed analysis in Section 4.4.3 and Appendix E.1, showing that many SSL methods can negatively impact model calibration, leading to unreliable uncertainty measures that affect AL performance.
>
> Additionally, we provide results below for a naive combination of SSL and AL on KITTI, further showcasing this conflict, which is often overlooked by existing SSL + AL frameworks.
>
> Tabel III: Results of naive combinations of SSL and AL on KITTI
>
> | Settings           | Car  | Ped  | Cyc  | Avg  |
> | ------------------ | ---- | ---- | ---- | ---- |
> | Normal+Entropy     | 73.6 | 48.2 | 51.9 | 57.9 |
> | 3DIouMatch+Entropy | 74.0 | 43.2 | 49.9 | 55.7 |
> | HSSDA+Entropy      | 74.9 | 41.7 | 52.5 | 56.4 |
> | Normal+CRB         | 73.3 | 45.3 | 47.4 | 55.3 |
> | 3DIouMatch+CRB     | 74.6 | 40.8 | 36.7 | 50.7 |
> | HSSDA+CRB          | 74.8 | 41.5 | 40.2 | 52.2 |
>
> These results demonstrate that directly combining SSL—whether using the SOTA HSSDA method or a simpler approach like 3DIoUMatch—with AL often leads to performance degradation, highlighting the need for a carefully designed framework like ours to effectively resolve this conflict.
>
> **Q3: About the figure.**
>
> A3: Thank you for your feedback on the figures. We appreciate your suggestion and will carefully revise and enhance the figures to improve their clarity and aesthetic quality in the final version of the paper.

---

> > ### Comment · Reviewer_Dsaa · 2024-11-22
> >
> > After carefully reviewing the comments from other reviewers, I still believe that the innovation of this paper is limited, and I will maintain my score.

---

> > > ### Author Response · Authors · 2024-11-22
> > > **Clarification of Our Contributions**
> > >
> > > We sincerely thank you for your thoughtful comments.
> > >
> > > It is rather unfortunate to witness months of hard work receiving some frustrating feedback. However, it seems there may have been some misunderstandings, and we feel that **certain aspects of our work may have been overlooked**.
> > >
> > > Our task focuses on the **SSAL framework**, and our comparisons are presented in the context of existing SSAL methods. **A key contribution** of our paper is identifying and resolving the **conflicts between SSL and AL**, which are critical but **often ignored by prior methods**. We believe that **raising awareness of this issue and providing a robust solution are substantial contributions**.
> > >
> > > Regarding the comparison with existing SSL methods, we have provided experiments demonstrating that **directly adopting existing SSL methods** (**including SS3D**) in the SSAL framework can be harmful. In contrast, our CPSP module is designed specifically to address the challenges in the SSAL framework and has proven to be successful. This demonstrates that we have addressed the **core issue** rather than **superficially adapting existing methods**. We believe this approach prioritizes **clarity and effectiveness**, in contrast to methods that rely on overly complex designs with minimal actual improvement.
> > >
> > > Additionally, it seems our **strong active learning methods**, which are **an equally significant contribution of our work**, may have been overlooked.
> > >
> > > As for the **improvement**, we believe our results represent a **substantial enhancement**, strongly demonstrating our significant success in the SSAL framework.
> > >
> > > Finally, we respectfully suggest that our work be evaluated in its entirety, rather than focusing narrowly on one module and ignoring the broader contributions of the framework.
> > >
> > > We hope this clarifies our contributions, and we sincerely thank you for your thoughtful review.
> > >
> > > Once again, we sincerely thank you for your time and thoughtful review.

---

### Official Review · Reviewer_wW9A · 2024-11-03

**Soundness:** 2
**Presentation:** 2
**Contribution:** 2
**Rating:** 5
**Confidence:** 3

**Summary:**

The paper introduces BC-SSAL (Bidirectional Collaborative Semi-Supervised Active Learning) to enhance 3D object detection in LiDAR-based systems. The framework addresses the challenge of annotation burden by effectively leveraging unlabeled data through a combination of semi-supervised learning (SSL) and active learning (AL). Overall, this paper is technically detailed and the experiments are quite comprehensive. However, most of the designs in the paper are combinations of existing work, lacking novel designs, and the improvements in the experiments are not significant.

**Strengths:**

1. The paper is well-structured, with a clear abstract, introduction, methodology, experiments, and conclusion sections that logically flow from one to the next.
2. The paper provides extensive experimental results on the KITTI and WOD datasets and conducts a large number of ablation studies.
3. The paper provides a multitude of figures that clearly demonstrate the design details of each module, facilitating the reader's understanding or reproduction of the methods described in the paper.

**Weaknesses:**

1. It is not clear why the method is named ‘Bidirectional xx’, I cannot see any module design that is ‘bidirectional’.
2. I cannot see any special contributions from this paper. Most of the components are simple combinations of existing works. Such as the so-called CPSP is indeed a GT sampling data augmentation used in most of regular 3D object detectors.
3. The improvement of this method does not seem to be very significant. On the KITTI dataset, the most convincing category is the car, as there are enough objects in this category to make the conclusions more reliable. However, this method is not even as good as 3DIoUMatch.

**Questions:**

1. Why is the paper titled 'Bidirectional xx'? Where is the bidirectional operation reflected?
2. What are the core differences between CPSP and GT sampling?

---

> ### Author Response · Authors · 2024-11-19
> **Response to Reviewer wW9A**
>
> Dear Reviewer wW9A,
>
> We sincerely appreciate your review with thoughtful comments. We have carefully considered each of your questions and provide detailed responses below to clarify any misunderstandings. Please let us know if you have any further questions or concerns.
>
> **Q1: The clarification about ‘Bidirectional Collaboration’.**
>
> A1: The design of our method is bidirectional, reflecting the two-way interaction between SSL and AL:
>
> - **From SSL to AL:** We analyze why previous SSL methods often fail to integrate well with AL. The conflicts primarily stem from learning with unconfident objects, which introduces noise and unreliable uncertainty measures. To address this, our approach selectively learns only from confident objects by generating pseudo-scenes from unlabeled data that include confident objects while excluding those with high uncertainty. Furthermore, our uncertainty measure is aligned with the AL methods, ensuring consistency and better integration.
> - **From AL to SSL:** The primary contribution lies in the integration of AL with our pre-trained CPSP model. One key challenge is that the CPSP pre-trained model may "forget" some original true and confident objects. To address this, we propose an ensemble strategy to provide more reliable uncertainty measures. Additionally, our approach tackles class imbalance, a critical issue in outdoor scenes.  Our design helps reduce FPs, enabling the model to sample objects from the correct classes more effectively. The importance of this can be further understood in Reviewer VC1w's Q2.
>
>  **Q2: The contributions and the difference between our CPSP and GT sampling.**
>
> A2: First, one of the key contributions of our paper is **identifying and addressing the conflict between AL and SSL**, a critical issue overlooked by many previous SSAL frameworks. Ignoring this conflict often leads to inferior performance, as shown in our analysis and experiments.
>
> Furthermore, we believe our modules are not simple combinations of existing works but are **specifically designed and carefully integrated** to effectively resolve these conflicts. Each component is tailored to address the challenges inherent in combining AL and SSL, ensuring they work collaboratively to achieve superior performance. For example, in the case of **CPSP**, the key contribution lies in **how we select suitable pseudo boxes for training** during iterations. Unlike standard GT sampling, which is limited to GT objects , CPSP introduces a more nuanced and dynamic selection process, enabling the effective use of pseudo-labels while minimizing noise and ensuring robust performance.
>
> **Q3: The clarification about improvement.**
>
> A3: The goal of our method is to **improve the overall mAP across all classes,** especially in challenging caterories like pedestrians and cyclists, and our method consistently outperforms others on both KITTI and Waymo datasets. Considering the number of objects, especially in Waymo, the test samples across all classes are sufficient to validate the effectiveness of our approach. We believe the average mAP improvement shown in Tables 1 and 2 is substantial (such as compared to 3DIoUMatch, our method bring an improvement 2.6% mAP on KITTI and 1.9% mAP on Waymo) , achieved by only changing the training data.
>
> Although 3DIoUMatch outperforms our method in the Car category on KITTI, this primarily stems from **class imbalance**. Specifically, in the setting with 350 labeled objects, 3DIoUMatch samples 264 Cars, 63 Pedestrians, and 21 Cyclists, while our method samples 181 Cars, 128 Pedestrians, and 38 Cyclists. This gives 3DIoUMatch an advantage in the Car category due to over-sampling but results in weaker performance for other classes. We believe their sampling distribution is not ideal for object detection tasks where all classes are important. When we adjusted our sampling strategy by assigning significantly higher weights to the Car class (~260 Cars), our method achieved an 81.6 mAP in Car_mod, surpassing 3DIoUMatch (80.8). This result demonstrates that our method can achieve competitive performance in individual categories while maintaining balanced performance across all classes.

---

> > ### Comment · Reviewer_wW9A · 2024-11-22
> > **Response to author**
> >
> > I have carefully read all the reviewers' comments and the authors' responses. And I also compared the CPSP proposed in this paper with SS3D, whose core design is very similar. Although the authors claim some differences, they are not fundamentally different. I felt that minor changes to existing work in similar areas and with marginal improvement would not meet the standards of a top conference, so I kept my original recommendation.

---

> > > ### Author Response · Authors · 2024-11-22
> > > **Clarification of Our Contributions**
> > >
> > > We sincerely thank you for your thoughtful comments.
> > >
> > > It is rather unfortunate to witness months of hard work receiving some frustrating feedback. However, it seems there may have been some misunderstandings, and we feel that **certain aspects of our work may have been overlooked**.
> > >
> > > Our task focuses on the **SSAL framework**, and our comparisons are presented in the context of existing SSAL methods. **A key contribution** of our paper is identifying and resolving the **conflicts between SSL and AL**, which are critical but **often ignored by prior methods**. We believe that **raising awareness of this issue and providing a robust solution are substantial contributions**.
> > >
> > > Regarding the comparison with existing SSL methods, we have provided experiments demonstrating that **directly adopting existing SSL methods** (**including SS3D**) in the SSAL framework can be harmful. In contrast, our CPSP module is designed specifically to address the challenges in the SSAL framework and has proven to be successful. This demonstrates that we have addressed the **core issue** rather than **superficially adapting existing methods**. We believe this approach prioritizes **clarity and effectiveness**, in contrast to methods that rely on overly complex designs with minimal actual improvement.
> > >
> > > Additionally, it seems our **strong active learning methods**, which are **an equally significant contribution of our work**, may have been overlooked.
> > >
> > > As for the **improvement**, we believe our results represent a **substantial enhancement** rather than the so-called "marginal improvement," strongly demonstrating our significant success in the SSAL framework.
> > >
> > > Finally, we respectfully suggest that our work be evaluated in its entirety, rather than focusing narrowly on one module and ignoring the broader contributions of the framework.
> > >
> > > We hope this clarifies our contributions, and we sincerely thank you for your thoughtful review.
> > >
> > > Once again, we sincerely thank you for your time and thoughtful review.

---

### Official Review · Reviewer_BLqE · 2024-11-03

**Soundness:** 2
**Presentation:** 2
**Contribution:** 2
**Rating:** 3
**Confidence:** 4

**Summary:**

The paper introduces a Bidirectional Collaborative Semi-Supervised Active Learning (BC-SSAL) framework for 3D object detection in LiDAR data, aiming to reduce the annotation burden. BC-SSAL integrates semi-supervised learning (SSL) with active learning (AL) to leverage unlabeled data effectively. Experiments on KITTI and Waymo datasets show BC-SSAL achieves state-of-the-art performance.

**Strengths:**

This method combines active learning with semi-supervised learning, proposing a new scheme for efficient label learning. The idea is interesting. The proposed method achieves the best performance compared to multiple baseline methods on two datasets.

**Weaknesses:**

1. The Pre-train method proposed in this paper has limited novelty. The main content highly overlaps with SS3D[1]. Specifically, SS3D combines a pre-trained detector to mine potential instances in unlabeled scenes to generate an 'Instance Bank' and 'Broken Scene', and then uses gt-sampling to produce training data.
2. The effectiveness of the active learning module is missing validation. One of the main contributions of this paper is active learning, so it is necessary to supplement a separate performance comparison with the state-of-the-art (SOTA) active learning methods [2][3].
3. The method is called "Bidirectional Collaborative Semi-Supervised Active Learning," but no clear design addresses this specific collaboration. It is recommended that the authors clarify the specific conflicts between the two strategies, provide experimental data to demonstrate the consequences of such conflicts, and explain how these conflicts are resolved.
4. The semi-supervised scheme only employs HSSDA, lacking validation across different semi-supervised approaches. It is suggested that the authors validate the proposed bidirectional collaboration scheme across multiple representative semi-supervised methods.

[1] SS3D: Sparsely-Supervised 3D Object Detection from Point Cloud. CVPR 2022
[2] KECOR: Kernel Coding Rate Maximization for Active 3D Object Detection. ICCV 2023
[3] Exploringactive 3d object detection from a generalization perspective.

**Questions:**

What are the differences between the design of the CPSP module and the related model design in SS3D? Please also respond to the comments in Weaknesses.

---

> ### Author Response · Authors · 2024-11-19
> **Response to Reviewer BLqE (Part 1)**
>
> Dear Reviewer BLqE,
>
> We sincerely appreciate your review with thoughtful comments. We have carefully considered each of your questions and provide detailed responses below to clarify any misunderstandings. Please let us know if you have any further questions or concerns.
>
> **Q1: The difference between our CPSP and SS3D[1].**
>
> A1:  While our CPSP shares some conceptual similarities with SS3D, it differs fundamentally in both design and purpose. The key distinction between our CPSP and existing SS3D lies in **how pseudo-objects are selected for training**. SS3D is designed for a sparse-supervised setting, focusing on extracting as many unlabeled objects as possible, whereas CPSP is specifically tailored to support the AL process. This fundamental difference leads SS3D to overlook two critical aspects that are central to CPSP: the **box bank deletion mechanism** and the **use of backgrounds likely to be false positives (FPs) for training**. Let’s delve into these two factors to illustrate their importance.
>
> - **Box Bank Deletion Mechanism:**
>
>   No matter how confident or refined the filtering process is, pseudo-labels in the box bank inevitably contain errors and noise. Without a mechanism to delete these noisy labels, they propagate throughout training, resulting in confirmation bias that adversely affects uncertainty measurement. CPSP incorporates a robust deletion mechanism to address this issue, ensuring cleaner pseudo-labels and better AL outcomes.
>
> - **Use of Backgrounds Likely to Be FPs:**
>
>   CPSP utilizes background regions that are likely to be FPs, providing **negative signals** to the model. These negative signals help the model avoid overconfidence in incorrect predictions, thereby improving the reliability of uncertainty measures.
>
> To further demonstrate these differences, we conducted experiments by directly applying SS3D  within our framework. The results, presented below, compare the mAP and D-ECE (calibration, where lower is better) metrics for various configurations:
>
> Table I: mAP comparison across settings on Waymo
>
> | Settings   | Veh           | Ped           | Cyc           | Avg           |
> | ---------- | ------------- | ------------- | ------------- | ------------- |
> | Normal+CAL | 62.2/54.3     | 61.7/53.0     | 55.4/53.5     | 59.8/53.6     |
> | SS3D+CAL   | 62.7/54.8     | 60.2/51.9     | 54.2/52.1     | 59.1/52.9     |
> | CPSP+CAL   | **62.8/54.8** | **62.8/54.1** | **57.3/55.3** | **61.0/54.7** |
>
> Table II: D-ECE comparison across different pre-training methods on Waymo
>
> | Pre-train | Veh      | Pe       | Cyc      |
> | --------- | -------- | -------- | -------- |
> | Normal    | 0.11     | 0.10     | 0.25     |
> | SS3D      | 0.11     | 0.16     | 0.38     |
> | CPSP      | **0.09** | **0.08** | **0.15** |
>
> The results indicate that SS3D negatively impacts calibration (Table II), particularly for harder classes (pedestrians and cyclists), thereby limiting its effectiveness in supporting the AL process (Table I). CPSP, on the other hand, demonstrates significant improvements in both performance and calibration, validating the necessity of its design choices.
>
> **Q2: Validation of our CAL module.**
>
> A2: In fact, the effectiveness of our active learning (CAL) module has already been demonstrated in various settings, as shown in Tables 1 and 2 in the paper, where it outperforms other active learning methods, including KECOR[2] and CRB[3].
>
> Additionally, we provide a direct comparison with CRB and KECOR below, further showcasing the superiority of our method.
>
> Tabel III: Performance comparison of AL on KITTI
>
> | Methods | Car  | Ped  | Cyc  | Avg_mod |
> | ------- | ---- | ---- | ---- | ------- |
> | CRB     | 73.3 | 45.3 | 47.4 | 55.3    |
> | KECOR   | 73.2 | 46.7 | 48.2 | 56.0    |
> | CAL     | 74.6 | 51.1 | 54.4 | 60.1    |
>
> Tabel IV: Performance comparison of AL on Waymo
>
> | Methods | Veh       | Ped       | Cyc       | Avg       |
> | ------- | --------- | --------- | --------- | --------- |
> | CRB     | 62.7/54.4 | 56.6/48.4 | 54.6/52.7 | 57.9/51.8 |
> | KECOR   | 61.8/53.5 | 57.1/49.0 | 52.1/50.2 | 57.0/50.8 |
> | CAL     | 61.3/53.4 | 60.5/51.9 | 52.3/50.6 | 58.1/52.0 |

---

> ### Author Response · Authors · 2024-11-19
> **Response to Reviewer BLqE (Part 2)**
>
> **Q3: Detailed explaination about conflicts and bidirectional collaboration.**
>
> A3.1 (Conflicts):
>
> **Clarification of the conflicts**: The conflicts primarily arise from learning with unconfident objects, which are characterized by high uncertainty. Assigning incorrect pseudo-labels to these objects in SSL introduces significant noise, making it difficult for AL to reliably assess their uncertainties.
>
> **Consequences of the conflicts:**  As demonstrated in Section 4.4.3 and Appendix E.1, previous SSL methods negatively impact model calibration, resulting in unreliable uncertainty measures that hinder AL performance. This is evident in Tables 1 and 2, where these SSL pre-training methods combined with AL perform worse than their counterparts (Normal + AL), proving that these conflicts significantly affect performance.
>
> **Solution to the conflicts:** Our solution is to selectively learn only from confident objects. Specifically, we generate pseudo-scenes from unlabeled data by including only confident objects while excluding those with high uncertainty. Pre-training on these pseudo-scenes ensures that unconfident objects are not influenced by their own noisy pseudo-labels, thereby significantly mitigating the negative impact of mislabeling and noise. As shown in Section 4.4.3 and Appendix E.1, our approach achieves better calibration across all classes. This improved calibration supports AL methods in achieving superior results, as demonstrated in Tables 1 and 2.
>
> A3.2 (Bidirectional Collaboration):
>
> The design of our method is bidirectional collaborative, reflecting the two-way interaction between SSL and AL:
>
> - **From SSL to AL:** We address the conflicts discussed above, ensuring SSL pre-training effectively supports AL by focusing on confident objects and aligning uncertainty measures with AL strategies.
> - **From AL to SSL:** The primary contribution lies in the integration of AL with our pre-trained CPSP model. One key challenge is that the CPSP pre-trained model may "forget" some original true and confident objects. To address this, we propose an ensemble strategy to provide more reliable uncertainty measures. Additionally, our approach tackles class imbalance, a critical issue in outdoor scenes.  Our design helps reduce FPs, enabling the model to sample objects from the correct classes more effectively. The importance of this can be further understood in Reviewer VC1w's Q2.
>
> **Q4: Validation across different semi-supervised schemes.**
>
> A4: Firstly, HSSDA is a state-of-the-art SSL method, and our success with it demonstrates the effectiveness of our approach. Secondly, we have already validated the proposed scheme across different SSL methods in the Final Model Delivering (FMD) stage: Table 1 uses HSSDA, Table 2 uses CPSP, and Table F in the appendix also uses CPSP. **These experiments provide strong evidence of our method’s effectiveness.** Lastly, conducting additional experiments with multiple SSL methods in the FMD stage is extremely time-consuming due to the combination of various SSL and AL methods we evaluated (e.g., Table 1 and Table 2). **Given resource constraints, we focused on the most representative settings.**

---

> ### Comment · Reviewer_BLqE · 2024-11-27
>
> Thank you for your response. I have carefully read the comments of the other reviewers as well as your reply. I agree with the majority of the reviewers that the proposed method is quite similar to existing methods, so I will maintain my rating.

---

> > ### Author Response · Authors · 2024-11-27
> > **Clarification of Our Contributions**
> >
> > Thank you for your feedback. While we understand the concern about perceived similarities to existing methods, we firmly assert that our work introduces **clear and novel contributions** that address critical challenges within the **SSAL framework**.
> >
> > ### **1. Overall Contributions**
> >
> > Our work provides a **comprehensive solution** to the **SSAL framework**, addressing the **critical conflict between SSL and AL**—a key issue overlooked by prior methods. This conflict arises from noisy pseudo-labels in SSL disrupting AL’s uncertainty measures. Our **bidirectional collaboration** directly resolves this issue by ensuring:
> >
> > - **SSL improves AL**: Our CPSP module incorporates a **box bank deletion mechanism** to reduce noise propagation and a **novel use of false-positive backgrounds** to deliver negative signals, enhancing model calibration and improving uncertainty estimation—critical for effective AL.
> > - **AL enhances SSL**: Our CAL strategy addresses **class imbalance**, ensuring targeted sampling of rare categories such as cyclists, while simultaneously improving the **reliability of uncertainty measures**.
> >
> > These mechanisms are tailored specifically for SSAL, overcoming limitations of existing methods like SS3D or REDB, which do not address these conflicts.
> >
> > ### **2. Collaborative Active Learning (CAL)**
> >
> > Our CAL strategy is a well-designed module that:
> >
> > - **Enhances uncertainty estimation**: By leveraging an ensemble approach to correct biases in pseudo-labels, CAL ensures robust and reliable sample selection.
> > - **Addresses class imbalance**: Unlike traditional methods, CAL directly targets rare categories while avoiding over-sampling of irrelevant or redundant examples, enabling **balanced and effective sampling across all classes**.
> >
> > ### **Why This Matters**
> >
> > These contributions are not incremental but solve **fundamental challenges** in SSAL. Our work achieves **substantial improvements**, as demonstrated by a **1.0% mAP increase on KITTI** and **1.2% on Waymo**—a significant leap in highly competitive benchmarks. Importantly, our improvements are consistent across rare and frequent categories, showcasing the robustness and practicality of our approach.
> >
> > We believe these clarifications reinforce the **novelty, significance, and impact** of our work. We respectfully request a reconsideration of our contributions within the broader context of addressing key SSAL challenges. Thank you again for your time and effort in reviewing our work.

---

### Official Review · Reviewer_VC1w · 2024-11-04

**Soundness:** 2
**Presentation:** 2
**Contribution:** 3
**Rating:** 5
**Confidence:** 4

**Summary:**

This paper aims to reduce the annotation burden in LiDAR-based 3D object detection, the authors propose a Bidirectional Collaborative Semi-Supervised Active Learning framework (BC-SSAL). This framework combines Collaborative PseudoScene Pre-training (CPSP) to effectively utilize unlabeled data and Collaborative Active Learning (CAL) to enhance sampling, particularly for rare classes, in outdoor LiDAR scenes.

**Strengths:**

- The task setting is practical, combining active learning (AL) and semi-supervised learning (SSL) to enhance 3D detection performance while minimizing annotation requirements.
- Good exploration and analysis are conducted on various strategies for integrating SSL with existing AL frameworks.

**Weaknesses:**

- The proposed Collaborative PseudoScene Pre-training (CPSP) module shares similarities with the approach in [A], in which high-quality, high-certainty bounding boxes are stored in a memory bank, and point clouds from these boxes are integrated into scenes. Could you clarify and compare the conceptual and empirical differences between CPSP and [A]?

- The Active Learning (AL) sampling strategy lacks novelty, as entropy is a widely used general AL method. Moreover, the box diversity metric shows minimal improvement, as demonstrated in Table 4.

- Some typos: In lines 177 and 179, there should be a space before "(TMU)" and "(USS)."


[A] Chen, Z., Luo, Y., Wang, Z., Baktashmotlagh, M., & Huang, Z. (2023). Revisiting domain-adaptive 3D object detection by reliable, diverse and class-balanced pseudo-labeling. In Proceedings of the IEEE/CVF International Conference on Computer Vision (pp. 3714-3726).

**Questions:**

The impact of varying thresholds for confident object extraction has not been studied. Given that the model was initially pretrained on a limited dataset, its predictions may contain noise, suggesting that thresholds should be carefully tuned and selected. Could you explain the process for determining the optimal thresholds and demonstrate the impact of different threshold values?

---

> ### Author Response · Authors · 2024-11-19
> **Response to Reviewer VC1w (Part 1)**
>
> Dear Reviewer VC1w,
>
> We sincerely appreciate your review with thoughtful comments. We have carefully considered each of your questions and provide detailed responses below. Please let us know if you have any further questions or concerns.
>
> **Q1: The difference between our CPSP and REDB [A].**
>
> A1: While our CPSP shares some conceptual similarities with REDB, it differs fundamentally in both design and purpose. The key distinction between our CPSP and REDB lies in **how pseudo-objects are selected for training**. REDB is designed for a cross-domain setting, focusing on identifying objects that remain stable across source and target domains. In contrast, CPSP is specifically tailored to support the AL process in a single-domain setting. This fundamental difference results in two critical distinctions that are central to CPSP: the **box bank deletion mechanism** and the **use of backgrounds likely to be false positives (FPs) for training**. Let’s delve into these two factors to illustrate their importance.
>
> - **Box Bank Deletion Mechanism:**
>
>   No matter how confident or refined the filtering process is, pseudo-labels in the box bank inevitably contain errors and noise. Without a mechanism to delete these noisy labels, they propagate throughout training, resulting in confirmation bias that adversely affects uncertainty measurement. CPSP incorporates a robust deletion mechanism to address this issue, ensuring cleaner pseudo-labels and better AL outcomes. In contrast,  REDB filters objects by inserting them into random sampling scenes (e.g., CDE). This approach, while reasonable in cross-domain scenarios, may not work effectively in single-domain settings. Randomly placing objects into unrelated scenes (e.g., a car in a forest) can generate unreasonable predictions, reducing the stability of pseudo-labels and potentially discarding true and confident objects during training.
>
> - **Use of Backgrounds Likely to Be FPs:**
>
>   CPSP utilizes background regions that are likely to be FPs, providing **negative signals** to the model. These negative signals help the model avoid overconfidence in incorrect predictions, thereby improving the reliability of uncertainty measures. Conversely, REDB lacks this design, which limits its effectiveness in supporting the AL process.
>
> To further demonstrate these differences, we conducted experiments by directly applying REDB and its CDE component into our framework with the same threshold. The results, presented below, compare the mAP and D-ECE (calibration, where lower is better) metrics for various configurations:
>
> Table I: mAP comparison across settings on Waymo
>
> | Settings   | Veh           | Ped           | Cyc           | Avg           |
> | ---------- | ------------- | ------------- | ------------- | ------------- |
> | Normal+CAL | 62.2/54.3     | 61.7/53.0     | 55.4/53.5     | 59.8/53.6     |
> | CDE + CAL  | 62.5/54.5     | 62.0/53.2     | 54.0/51.9     | 59.5/53.2     |
> | REDB + CAL | **63.1/55.0** | 61.2/52.1     | 55.8/54.0     | 60.1/53.7     |
> | CPSP+CAL   | 62.8/54.8     | **62.8/54.1** | **57.3/55.3** | **61.0/54.7** |
>
> Table II: D-ECE comparison across different pre-training methods on Waymo
>
> | Pre-train | Veh      | Ped      | Cyc      |
> | --------- | -------- | -------- | -------- |
> | Normal    | 0.11     | 0.10     | 0.25     |
> | CDE       | 0.09     | 0.21     | 0.33     |
> | REDB      | **0.08** | 0.25     | 0.31     |
> | CPSP      | 0.09     | **0.08** | **0.15** |
>
> The results indicate that while CDE provides some improvements in calibration for vehicles, it fails to improve calibration for the more challenging classes, pedestrians and cyclists (Table II), thereby limiting its overall effectiveness in supporting the AL process (Table I).
>
> Finally, we acknowledge that some modules from REDB, such as OBC, may be beneficial for better training of objects. We will consider incorporating such modules into our framework in future work.

---

> ### Author Response · Authors · 2024-11-19
> **Response to Reviewer VC1w (Part 2)**
>
> **Q2: The Novelty of our CAL.**
>
> A2: The primary novelty of our CAL lies in **its integration with our pre-trained CPSP model**. While we acknowledge that entropy is a widely used metric, our approach goes beyond its simple application. One key challenge is that the CPSP pre-trained model can "forget" some original true and confident objects. To address this, we propose an **ensemble strategy**, ensuring more reliable uncertainty measures. Moreover, our AL method is highly flexible and can easily incorporate future advancements in uncertainty estimation. Additionally, our approach effectively addresses **class imbalance**, a critical issue often underestimated in outdoor scenes. The co-occurrence of multiple objects in a scene complicates the selection process, and FPs can result in sampling irrelevant scenes that lack the intended objects, particularly for rare classes like cyclists. This challenge is further exacerbated by many pre-trained models that generate excessive FPs. Unlike previous methods, our class balance strategy explicitly tackles these issues, enabling more effective and balanced sampling across all classes.
>
> **Q3: Typos.**
>
> A3: We appreciate your suggestions for improving the presentation of our paper. We will revise our paper carefully.
>
> **Q4: About the thresholds for confident object extraction.**
>
> A4: This is a great question! Noise is indeed a major reason why many SSL methods struggle with calibration and performance. Initially, we considered setting a high confidence threshold, but score distributions vary significantly across classes and models, making it difficult to determine a universally appropriate threshold. As a result, we abandoned this approach. We also explored top-k selection; however, this method can easily introduce noise and requires careful tuning of the value k, which is also challenging to determine.
>
> Instead, we opted for a **score clustering method**, setting a high number of clusters (KITTI: 20, Waymo: 50, which we found to be effective compared to fewer clusters (<10) or excessive clusters (>100)). By extracting objects from the highest-confidence cluster, this approach effectively minimizes noise while providing a sufficient number of pseudo-objects for training.

---

> ### Comment · Reviewer_VC1w · 2024-11-26
>
> Thanks for your thorough response. Most of my concerns have been addressed.

---

> > ### Author Response · Authors · 2024-11-26
> > **Thanks for Your Response**
> >
> > **Thank you for reviewing our paper.** If you feel that our response has addressed your concerns, we kindly ask you to consider adjusting your scores accordingly. If you have any further questions or concerns, please do not hesitate to let us know. We are happy to provide additional clarifications if needed.

---

### Author Response · Authors · 2024-11-22
**Response Reminder**

We have taken your initial feedback into careful consideration and have provided our responses accordingly. Would you mind confirming whether our responses have satisfactorily addressed your concerns? If you believe we have successfully addressed your concerns, we kindly request that you consider adjusting your initial score accordingly. We understand that you are busy, and we would be very grateful if you could keep our response in mind during discussions with the AC and other reviewers. Please feel free to share any additional comments you may have.

Once again, we sincerely thank you for dedicating your time and effort to reviewing our work.

---

### Author Response · Authors · 2024-11-25
**Reviewer-Author Discussion Period Ends in TWO Days**

Thanks again for reviewing our paper. We hope that our response adequately addressed your concerns. As the deadline approaches, please let us know if you have any further questions before the reviewer-author discussion period ends. We understand your busy schedule, and we would greatly appreciate it if you could consider our response during discussions with the AC and other reviewers.

We are more than happy to address any additional concerns or questions you may have.

Once again, we sincerely thank you for dedicating your time and effort to reviewing our work.

---

### Author Response · Authors · 2024-11-28
**Emphasizing Once Again the Overall Contributions of Our Work**

We have noticed that some reviewers, due to the similarity of one module in our paper to existing methods, **have overlooked the overall contributions** of our work. However, we would like to clarify the following points:

1. **The central contribution of our paper** is the **resolution of the conflict between SSL and AL** in the SSAL framework, a challenge that has been largely overlooked by prior methods. By resolving this conflict, we propose a **bidirectional collaboration** where SSL improves AL through calibrated pseudo-labels, and AL enhances SSL by addressing **class imbalance** and improving **uncertainty estimation**. This conflict resolution is fundamental to the success of SSAL, and it **differentiates our work** from previous SSAL frameworks.
2. **Our CPSP module is not a direct replication** of existing methods. It introduces **novel design choices**, such as the **box bank deletion mechanism** and the **use of false-positive backgrounds**, which improve **model calibration** and **uncertainty measures**—features not present in prior work like SS3D. These unique contributions are essential to making SSAL work effectively.
3. Our **CAL module** makes significant contributions by addressing **class imbalance**, improving **uncertainty estimation**, and eliminating redundant samples using **similarity metrics**. CAL ensures more **effective sample selection**, particularly for rare categories like cyclists, which are often neglected in other frameworks.

Overall, our work provides **a comprehensive and impactful solution** to the SSL-AL conflict in SSAL, with **important implications** for future research and application in the field. We respectfully ask the reviewers and AC to evaluate the **overall contributions** of our work, rather than focusing on the similarity of one module. Dismissing the **holistic value** of our approach would be an unfair evaluation, and we hope that the **entirety of our contributions** will be taken into account during the review process.

Thank you for your time and feedback.

---

### Author Response · Authors · 2024-12-02
**Response Reminder: Approaching Deadline**

**Thank you for reviewing our paper.** As the deadline is approaching, if our response has addressed your concerns, we kindly ask you to consider adjusting your scores accordingly. If you have any further questions or concerns, please let us know, and we will be happy to provide additional clarifications. Your feedback is greatly appreciated, and we look forward to your response before the reviewer-author discussion period ends.

---

### Meta-Review · Area_Chair_wrR9 · 2024-12-09

**Metareview:**

This paper investigates a unified framework integrating semi-supervised learning (SSL) and active learning (AL) to provide a cost-effective solution for 3D object detection. By employing a nested refinement loop, the AL component selects instances that are both uncertain and diverse, while the SSL component self-trains on high-confidence instances, thereby reducing annotation overhead. Additionally, class-balanced learning is incorporated, and the proposed approach is evaluated on two widely used benchmark datasets.

The paper’s strengths lie in the extensive experimental comparisons with numerous existing AL methods, demonstrating notable performance improvements across different classes. The results show performance improvements over numerous existing AL solutions, although these gains are somewhat expected due to the additional use of unlabeled data, which is not a typical consideration in mainstream AL methods.

The primary concern raised by all reviewers is that the paper’s core technical contribution remains unclear. While the integration of AL and SSL is relatively underexplored, the chosen methods for measuring uncertainty and diversity seem quite standard, and the overall architecture appears to be a high-level combination of existing approaches. To strengthen this work in the future, it would be more beneficial to target a more specific technical challenge in AL and SSL (such as, but definitely not limited to, how to adaptively control the annotation budget, how to set the threshold for SSL, or preventing forgetting during iterative training without excessive data resampling) and this can help audience to better understand the core innovation and facilitate future work in this fields. It is recommended authors carefully go through the reviewers' suggestions and reposition the focus of this work, which would help strengthen the quality and impact of this work.

**Additional Comments On Reviewer Discussion:**

During the rebuttal period, four reviewers (VC1w, BLqE, wW9A, and Dsaa) raised several concerns that the authors attempted to address through clarifications, additional experiments, and detailed explanations.

- Novelty Issue. Multiple reviewers found that the proposed Collaborative PseudoScene Pre-training (CPSP) module closely resembled components of existing methods (e.g., REDB and SS3D). They questioned whether the differences the authors highlighted for the proposed modules constituted true innovations rather than incremental adaptations. The authors responded by providing detailed comparisons and additional experiments, showing that directly substituting their module with REDB or SS3D led to poorer calibration and performance. Reviewers BLqE, wW9A, and Dsaa remained unconvinced that the modifications represented a substantial conceptual advance over prior work.
- "Bidirectional". Reviewers BLqE, wW9A, Dsaa questioned the claim of “bidirectional” collaboration between SSL and AL, arguing that the paper did not clearly demonstrate a two-way interplay or a specialized mechanism reflecting this concept. The authors clarified that the bidirectionality lay in SSL improving calibration for AL, and AL helping refine SSL results by focusing on more confident samples and addressing class imbalance. Reviewers found the presentation still too high-level and not sufficiently distinct from standard SSL+AL combinations.

In forming a final decision, the positive aspects were the authors’ thorough and detailed rebuttal, additional comparisons, and clarifications.  The majority of reviewers (BLqE, wW9A, and Dsaa) remained unconvinced about the paper’s core innovation and the need for a dedicated "bidirectional" framework. They noted that, despite added explanations, the approach still appeared as a combination of existing components without a standout technical breakthrough. These reviewers maintained their initial assessments.

---

### Decision · Program_Chairs · 2025-01-22

Reject